# Precursor engineering of hydrotalcite-derived redox sorbents for reversible and stable thermochemical oxygen storage

Michael High[1,6], Clemens F. Patzschke[1,6], Liya Zheng[1,6], Dewang Zeng[1,2], Oriol Gavalda-Diaz[3,4], Nan Ding[1], Ka Ho Horace Chien[1], Zili Zhang[1], George E. Wilson[3], Andrey V. Berenov[3], Stephen J. Skinner[3], Kyra L. Sedransk Campbell[5], Rui Xiao[2]✉, Paul S. Fennell[1]✉ & Qilei Song[1]✉

Chemical looping processes based on multiple-step reduction and oxidation of metal oxides hold great promise for a variety of energy applications, such as $CO_2$ capture and conversion, gas separation, energy storage, and redox catalytic processes. Copper-based mixed oxides are one of the most promising candidate materials with a high oxygen storage capacity. However, the structural deterioration and sintering at high temperatures is one key scientific challenge. Herein, we report a precursor engineering approach to prepare durable copper-based redox sorbents for use in thermochemical looping processes for combustion and gas purification. Calcination of the CuMgAl hydrotalcite precursors formed mixed metal oxides consisting of CuO nanoparticles dispersed in the Mg-Al oxide support which inhibited the formation of copper aluminates during redox cycling. The copper-based redox sorbents demonstrated enhanced reaction rates, stable $O_2$ storage capacity over 500 redox cycles at 900 °C, and efficient gas purification over a broad temperature range. We expect that our materials design strategy has broad implications on synthesis and engineering of mixed metal oxides for a range of thermochemical processes and redox catalytic applications.

Anthropogenic emissions of $CO_2$ generated from the burning of fossil fuels are a primary driver for global warming and climate change. Besides the large-scale deployment of renewable energy and increased energy end-use efficiencies, $CO_2$ capture, utilization, and storage (CCUS) technologies are likely to be essential for mitigating industrial $CO_2$ emissions to an extent that allows the ambitious global Net-Zero emissions to be fulfilled. Amongst many emerging CCUS technologies, chemical looping processes have attracted significant attention for

energy-efficient conversion of fuels with inherent $CO_2$ capture[1–3]. Chemical looping processes involve the use of oxygen storage materials, commonly derived from metal oxides, which serve to transport oxygen through cyclic reduction and oxidation reactions (Fig. 1a)[2]. This concept has been extended to a variety of chemical processes, including chemical looping combustion (CLC)[4,5], methane reforming and partial oxidation[6–8], $CO_2$ conversion[9–11], hydrogen production[12,13], air separation[14–17], and redox catalytic processes for chemical

[1]Department of Chemical Engineering, Imperial College London, Exhibition Road, London SW7 2AZ, UK. [2]Key Laboratory of Energy Thermal Conversion and Control (Ministry of Education), School of Energy and Environment, Southeast University, Nanjing 210096, P.R. China. [3]Department of Materials, Imperial College London, Exhibition Road, London SW7 2AZ, UK. [4]Composites Research Group, University of Nottingham, Jubilee Campus, Nottingham NG7 2GX, UK. [5]Department of Chemical and Biological Engineering, The University of Sheffield, Western Bank, Sheffield S10 2TN, UK. [6]These authors contributed equally: Michael High, Clemens F. Patzschke, Liya Zheng. ✉e-mail: ruixiao@seu.edu.cn; p.fennell@imperial.ac.uk; q.song@imperial.ac.uk

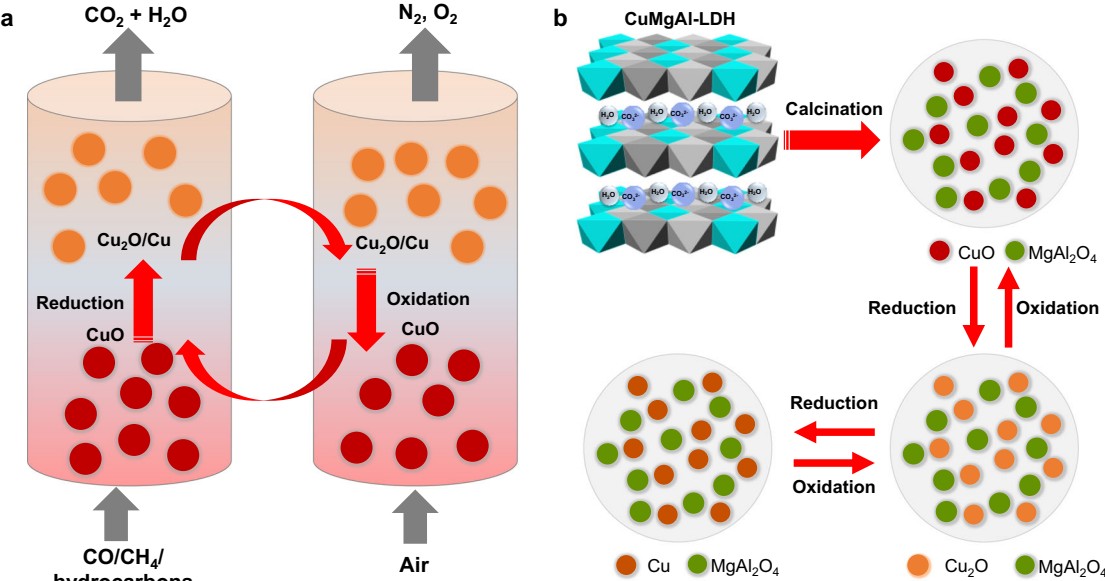

**Fig. 1 | Schematic illustration of thermochemical looping process and design strategy of oxygen storage materials. a** Schematic diagram showing the chemical looping redox cycle process in which copper oxides undergo cyclic reduction and oxidation reactions. The thermochemical redox reactions lead to a separate pure stream of $CO_2$ after water is removed, which can be utilized or stored readily.

**b** Synthesis of CuMgAl layered double hydroxides (LDHs) and derived CuMgAl mixed metal oxides (MMOs) consisting of $CuO/MgAl_2O_4$, in which the Mg-Al phase stabilizes the Cu phases from sintering and inhibits the formation of Cu-Al oxides during redox cycling between CuO, $Cu_2O$, and Cu.

conversions[18–20]. In the classical CLC process, metal oxides oxidize (upon reduction) a fuel (e.g. natural gas, coal, or biomass) in the fuel reactor. The exhaust stream, primarily $CO_2$ and $H_2O$, is then purified to (mainly) $CO_2$, e.g. by condensing the $H_2O$. The reduced metal oxides are then transported back to the air reactor, where they are re-oxidized in air. Deployment of CLC with solid fuels and conventional oxygen storage materials typically results in relatively slow reaction rates owing to the nature of solid-solid reactions. Therefore, chemical looping with oxygen uncoupling (CLOU), a variant of CLC, has been developed to overcome this problem[21]. This technology uses gas-phase $O_2$, generated by metal oxides such as those of copper, to react with the solid fuel directly, through comparatively fast solid-gas reactions. In the past two decades, pilot megawatt-scale CLC reactors have been developed[22–25], showing great promise for industrial scale decarbonization.

Chemical looping redox cycles are also promising for industrial gas purification processes. One application is argon recycling in the solar photovoltaic manufacturing industry, where a significant amount of high-purity argon gas is used in the fabrication of silicon wafers for solar panels. The exhaust gas stream usually contains contaminated gases such as CO and $H_2$ at several thousand ppm along with traces of other combustible gases. Chemical looping combustive purification (CLCP) has been developed to effectively remove these contaminants to below ppm levels. The metal oxides oxidize the combustible gases to $CO_2$ and $H_2O$, which are subsequently removed by molecular sieve sorbents. The reduced oxygen sorbents are then regenerated by oxidation in air. The CLCP has been successfully scaled up for argon or helium recovery and purification in industry[26]. In broad context, this principle can be applied to replace conventional regenerative thermal oxidizers operated at high temperature for clean-up of low-concentration hydrocarbons such as methane from landfills or volatile organic compounds (VOCs) at lower temperatures.

A common scientific challenge in these chemical looping processes is the development of high-performance sintering-resistant oxygen storage materials that can reversibly uptake and release oxygen over multiple redox cycles[5]. Perovskite-type oxides are

promising materials. However, their oxygen storage capacity is usually low[14,15,27–29]. Cu-based mixed oxides have been studied most extensively owing to their high oxygen capacity ($CuO/Cu$: 0.201 $g_{O2}/g_{CuO}$; $CuO/Cu_2O$: 0.101 $g_{O2}/g_{CuO}$), fast rates of reaction, and absence of thermodynamic limitations for the complete combustion of fuels. However, unsupported metal oxides are prone to sintering, and have a tendency to agglomerate during operation owing to the low Tammann temperatures of the copper oxides (CuO: 526 °C, $Cu_2O$: 481 °C)[30]. To increase their mechanical strength, the copper oxides are often supported on $SiO_2$[31] or $Al_2O_3$[32]. However, the fragmentation and agglomeration of these materials led to performance degradation[31]. Furthermore, over extended periods of redox cycling, the CuO reacts with $Al_2O_3$ to form the $CuAl_2O_4$ spinel oxides[33]. At typical CLC/CLOU operating temperatures, $CuAl_2O_4$ has a lower $O_2$ equilibrium partial pressure, slower $O_2$ release rates, and a lower $O_2$ release capacity than CuO[34]. Thus, it is desirable to develop a material that inhibits formation of $CuAl_2O_4$. Co-precipitation has been used to prepare Cu-Al mixed oxides[35], however, the fundamental understanding of the precursor chemistry is usually lacking.

Although significant efforts have been devoted to the development of Cu-based oxides (see summary in Supplementary Table S1)[36–39], an alternative approach is desirable to control the synthetic chemistry of precursors for improved long-term reactivity and stability of their derived mixed metal oxides (MMOs). We previously reported the synthesis of hydrotalcite-like (or layered double hydroxides, LDHs) precursors and derived MMOs for chemical looping processes[40]. Although LDH chemistry and the derived MMOs have been well studied for decades, especially for catalysis applications[41–43], it is still an innovative approach to use the LDH-derived materials as oxygen storage materials for high-temperature chemical looping processes. Owing to the high atomic level dispersion of elements in the LDH precursors, the resulting mixed oxides consist of a highly dispersed phase within the support and showed fast oxygen release rates and redox stability. Furthermore, we found that the residual alkaline species (sodium or potassium) effectively stabilize the mixed oxides, as proven by other doping approaches[44,45]. However, it remains

challenging to precisely control the content of sodium-containing species which originate from the precipitation agent used for co-precipitation. Evaporation of the sodium at high temperatures will result in a loss of this stabilizing agent, which might deteriorate the materials over extended redox cycles. During long-term thermo-chemical redox-cycling, sodium tends to react with $Al_2O_3$ and form a variety of species (i.e. $Na_xAl_yO_z$)[40]. It is desirable to tune the synthetic chemistry of the LDH precursors to minimize the formation of sodium-containing impurities and achieve a more stable long-term cycling performance.

In this work, we report the precursor engineering of CuMgAl-LDHs to derive Cu-based MMOs and demonstrated their applications as high-performance oxygen storage materials with high thermal stability and resistance to sintering over extended redox cycles. The CuMgAl-LDH precursors were synthesized by co-precipitation, during which the presence of Mg species inhibited the formation of sodium-containing species, which resulted in MMOs with negligible residual sodium contents. Owing to the homogeneous distribution of metal cations, thermal decomposition of the LDH precursors resulted in the formation of highly dispersed MMOs, consisting of a high loading of active CuO dispersed in the Mg-Al spinel oxide support (Fig. 1b). Since the spinel phase remained inert with respect to CuO, it was effective in inhibiting the formation of $CuAl_2O_4$ (Supplementary Fig. S1). The resulting MMOs demonstrated fast reaction rates, high oxygen capacity, and excellent stability against sintering over multiple cycles of reduction and oxidation between $CuO-Cu_2O-Cu$ in both a thermogravimetric analyser (TGA) and a fluidised bed reactor (FBR). The improved understanding of the precursor chemistry was used for further improvement of the materials and may be used for the development of new oxygen sorbents. Our material design strategy provides an alternative approach to the synthesis of oxygen storage materials and redox catalysts with promising applications in chemical looping processes, gas purification, thermochemical energy storage, and catalytic processes.

## Results

### Synthesis of LDH precursors and mixed oxides

A series of CuMgAl-LDH precursors were synthesized via co-precipitation using an aqueous solution consisting of a mixture of 1 M NaOH and 1 M $Na_2CO_3$ as the precipitating agent (details of the procedure are provided in the Methods section and in Supplementary Fig. S1). The synthetic chemistry was first tuned by varying the Mg/(Cu+Al) molar ratio between zero and 0.2, while keeping the total cation concentration constant at 2 M. Post-synthesis characterization of all materials before and after thermal decomposition (calcination) was performed and the results are presented in Fig. 2 and Supplementary Fig. S2.

To understand the structure and properties of the LDH precursors, we performed a range of characterization analyses. X-ray diffraction (XRD) patterns of the precursors (Fig. 2a) confirmed the formation of LDH structures with the observation of characteristic peaks of LDHs, for example, the (003), (006), and (009) peaks at Bragg angles (2θ) of 12°, 24°, and 35°, respectively. A comparison with references (drawn under the measured patterns) indicated that the structure is a hybrid between Cu-Al hydrotalcite and Mg-Al hydrotalcite, with the peaks matching best with CuAl hydrotalcite (JCPDS 46-0099), $Cu_6Al_2(OH)_{16}CO_3 \cdot 4 H_2O$ (JCPDS 37-0630) and $Mg_6Al_2(OH)_{16}CO_3 \cdot 4 H_2O$ (JCPDS 22-0700). This finding aligns well with the fact that a peak shift to lower Bragg angles occurred with an increasing molar Mg concentration – i.e. the material resembles decreasingly the CuAl LDH and increasingly the MgAl LDH. Fourier-transform infrared spectroscopy (FTIR) confirmed the LDH precursors and interlayer species (Supplementary Fig. S3). After calcination at 950 °C for 3 h in air, the active phase of the resulting MMOs consisted of CuO (JCPDS 80-1268) as analysed by XRD (Fig. 2b, Supplementary Figs. S2 and S4).

Additionally, the samples with a molar composition of 3:x:2 with $x \geq 0.2$ also showed peaks at positions corresponding to those of the reference for $MgAl_2O_4$ (JCPDS 73-1959), with their peak intensity increasing with an increase in the molar concentration of Mg. When increasing the Mg loading, the crystallite size of $MgAl_2O_4$ also increased steadily from $16.6 \pm 1.7$ nm to $43.9 \pm 8.8$ nm, while the crystallite size of CuO remained relatively constant in the range 37–54 nm. Detailed results of the crystallite size calculation are presented in Supplementary Table S2. The XRD patterns of the freshly calcined samples were compared with the reference pattern for $CuAl_2O_4$ (JCPDS 00-33-0448), which confirmed that this synthesis route was effective in inhibiting the unfavorable formation of this spinel oxide in all samples.

To study the composition of the LDH precursors and calcined oxides, XRF and ICP analysis were performed to quantify the content of Na and Mg. The full results of the ICP and XRF analyses are presented in the Supplementary Tables S3 and S4. Figure 2c presents the mass fractions of Na and Mg in the LDH and MMO samples, which were converted to mass fractions of the oxides (using stoichiometry and the relative molecular masses of the compounds) for better comparison with our previous work[40]. In the previous work on CuAl LDHs, Na-containing species such as dawsonite ($NaAlCO_3(OH)_2$) were formed when the co-precipitation was performed using $Na_2CO_3$ solution as a precipitating agent[40]. The calcination of dawsonite and formation of sodium aluminates ($NaAlO_2$ and other phases) inhibited the formation of $CuAl_2O_4$. As shown in Fig. 2c, the $Na_2O$ content in the LDH precursors and derived MMOs decreases with an increase in the loading of Mg. In the absence of Mg, significant amounts of Na species ($XRF_{Na_2O}$: 3 wt%) formed dawsonite during the co-precipitation of the CuAl LDH as reported in our previous work[40]. The sodium-rich phases persisted during calcination and inhibited the undesired solid-state reaction of CuO and $Al_2O_3$ to $CuAl_2O_4$. With an increase in the mass fraction of Mg the measured mass fraction of $Na_2O$ decreased. To test if the inhibition of sodium contamination is also effective at higher Cu-loadings, two further samples were prepared. In the three samples with a Cu:Mg:Al molar ratio of x:1:2 with x = 3, 4 and 5, almost no sodium was detected by both, inductively coupled plasma mass spectrometry (ICP-MS) and X-ray fluorescence (XRF): $ICP_{LDH} \leq 0.07$ wt%; $XRF_{LDH}$: ≤ 0.01 wt%, $XRF_{MMO}$: = 0.00 wt%. This indicates that the introduction of Mg is effective in inhibiting the formation of sodium-containing impurities over a larger Cu-loading range. A discussion on the mechanisms of inhibition of sodium-containing species by the presence of Mg has been given in the Supplementary Information. Thermodynamic calculations (Supplementary Fig. S5, Table S5) indicate that the presence of Mg suppresses thermodynamically the dawsonite formation and favours intermediate formation required for MgAl hydrotalcite formation (e.g. $MgCO_3$ and $Al(OH)_3$) at moderate pH values (i.e. pH 6.0–9.5). At higher pH values (pH > 9.5), significant Mg-Al hydrotalcite formation was predicted (log conc. > 0.1). Though, dawsonite is predicted to form over a wide pH value range, its equilibrium concentration is significantly lower when Mg is present. This thermodynamic suppression and fast kinetics of the formation of the hydrotalcite and its intermediates are likely to effectively inhibit dawsonite formation. Other sodium-containing species remain dissolved or are soluble, which allows their removal from the system upon washing. Thermodynamic calculations of MMOs (Supplementary Fig. S6) for the system Cu-Mg-Al-O at 600–1000 °C confirmed that $MgAl_2O_4$ forms and remains stable while the formation of $CuAl_2O_4$ and $CuAlO_2$ is inhibited. These fundamental understandings allowed us to synthesize LDH precursors and MMOs in a reproducible way and to scale up their synthesis to kilogram scale (Fig. 2d).

To understand the morphological and structural evolution, we characterized the LDH precursors and calcined mixed oxides with electron microscopy techniques. The SEM image of the CuMgAl-LDH precursor (Fig. 2e, Supplementary Fig. S7) showed a well-defined platelet-like structure with the longer dimension of the platelets

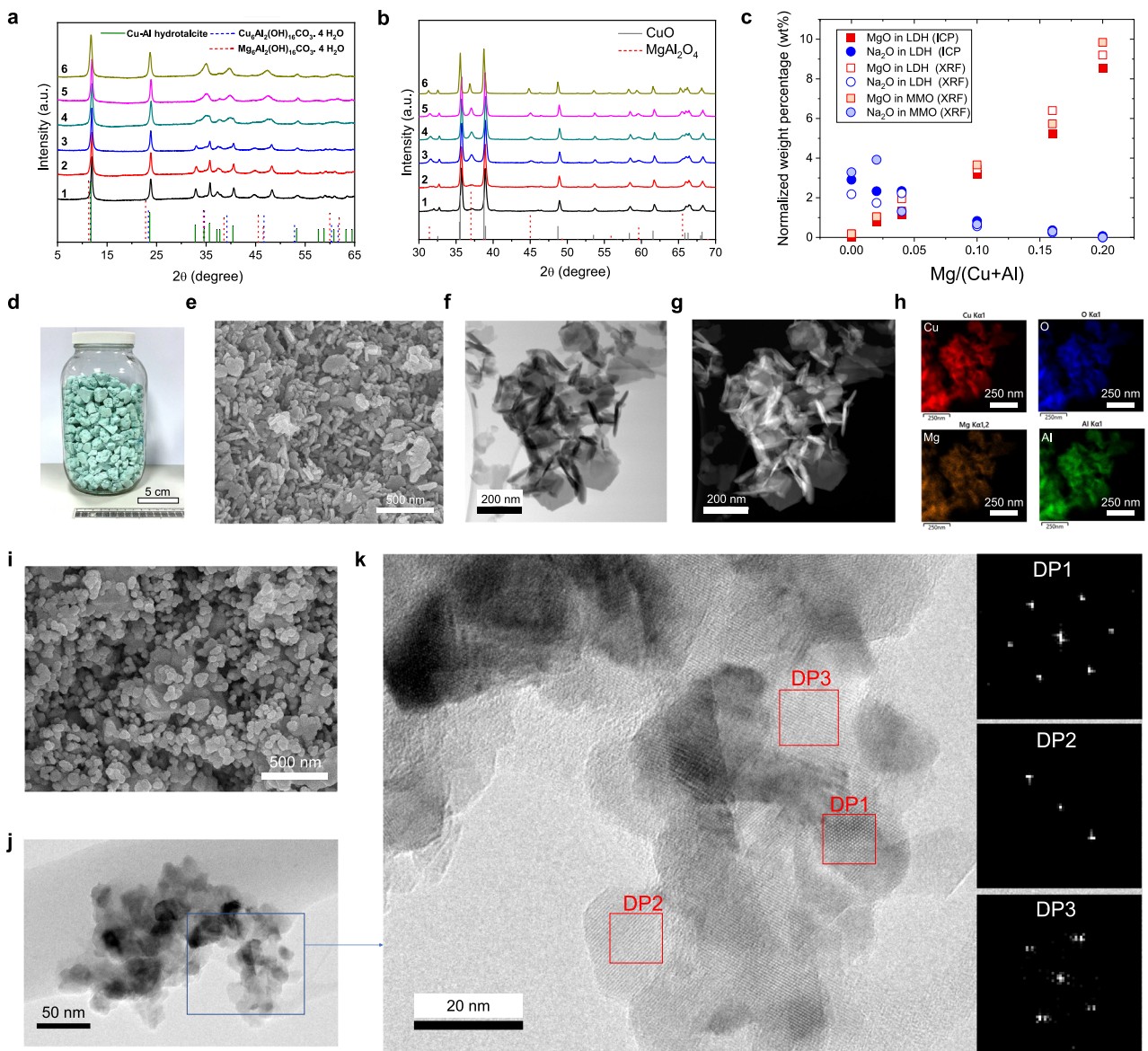

**Fig. 2 | Synthesis and characterization of CuMgAl-LDH precursors and derived MMOs. a** XRD patterns of CuMgAl-LDHs and **b**, CuMgAl-MMO. The crystalline pattern in **a** corresponds to CuAl hydrotalcite (JCPDS 46-0099, 37-0360; vertical lines), and MgAl hydrotalcite Mg₆Al₂(OH)₁₆CO₃·4 H₂O (JCPDS 22-0700; vertical dashed lines). Crystalline patterns in **b** correspond to CuO (JCPDS 80-1268; vertical lines) and MgAl₂O₄ (JCPDS 73-1959, vertical dashed lines), respectively. The apparent Cu:Mg:Al molar ratio for samples in **a** and **b**: (1) 3:0:2, (2) 3:0.1:2, (3) 3:0.2:2, (4) 3:0.5:2, (5) 3:0.8:2, (6) 3:1:2. **c** normalized weight percentage of Na₂O and MgO in the LDH precursor and calcined MMO, calculated from ICP and XRF measurements. **d** Photo and **e** SEM image of CuMgAl-LDH with a molar Cu:Mg:Al ratio of 3:1:2. **f** STEM and **g** HAADF-STEM images of CuMgAl-LDH precursor. **h** element mapping of CuMgAl-LDH precursor, Cu (red), Al (green), Mg (yellow), and O (blue). **i** SEM image of CuMgAl-MMO calcined at 950 °C. **j** STEM image of the calcined CuMgAl-MMO at 800 °C for 3 h. **k** HR-TEM image of CuMgAl-MMO calcined at 800 °C for 3 h with insets showing the fast Fourier transform (FFT) patterns of different regions.

measuring on average 110 ± 46 nm (the interval denotes the standard deviation when analysing 120 randomly selected platelets). Correspondingly, the average measured size for the shorter dimension of the platelets (i.e. their thickness) was 24.8 ± 4.8 nm. The scanning transmission electron microscopy (STEM), high-angle annular dark-field (HAADF)-STEM, and energy dispersive spectroscopy (EDS) analysis (Fig. 2f, g, and h) confirmed the platelet-like structure and uniform elemental distribution in the LDH precursors. Upon calcination, the platelets coalesced to larger more spherical platelets, which partially have developed grain-like features (Fig. 2i). TEM images of the MMOs calcined at a temperature of 800 °C show highly dispersed CuO particles (size smaller than 10 nm) in the support (Supplementary S8). Upon calcination at 800 °C for 3 h, the CuO particles grew to about 10–20 nm with a high degree of dispersion of the phases being

maintained (Fig. 2j, k). The fast Fourier transform (FFT) patterns of different regions clearly suggest the formation of crystalline domains, however specific assignment of crystalline phases could not be obtained, which might be due to defects in the Cu phases, interactions between Cu and the Mg-Al oxide support at their interfaces or partial incorporation of Cu into the support.

To further investigate the influence of Mg on the composition of the MMOs, we also synthesized control samples, including CuMgAl-LDH using an aqueous solution of 2 M NaOH as the precipitating agent, and performed detailed characterisation analyses (Supplementary Figs. S2, S3, S9–14, Tables S3–4, Table S6). The results confirmed that a significant amount of CuAl₂O₄ was formed in the CuAl-MMO, while addition of Mg (i.e. CuMgAl-LDHs and their MMOs) limited the formation of CuAl₂O₄ (Supplementary Figs. S2 and S4, Table S4). FTIR

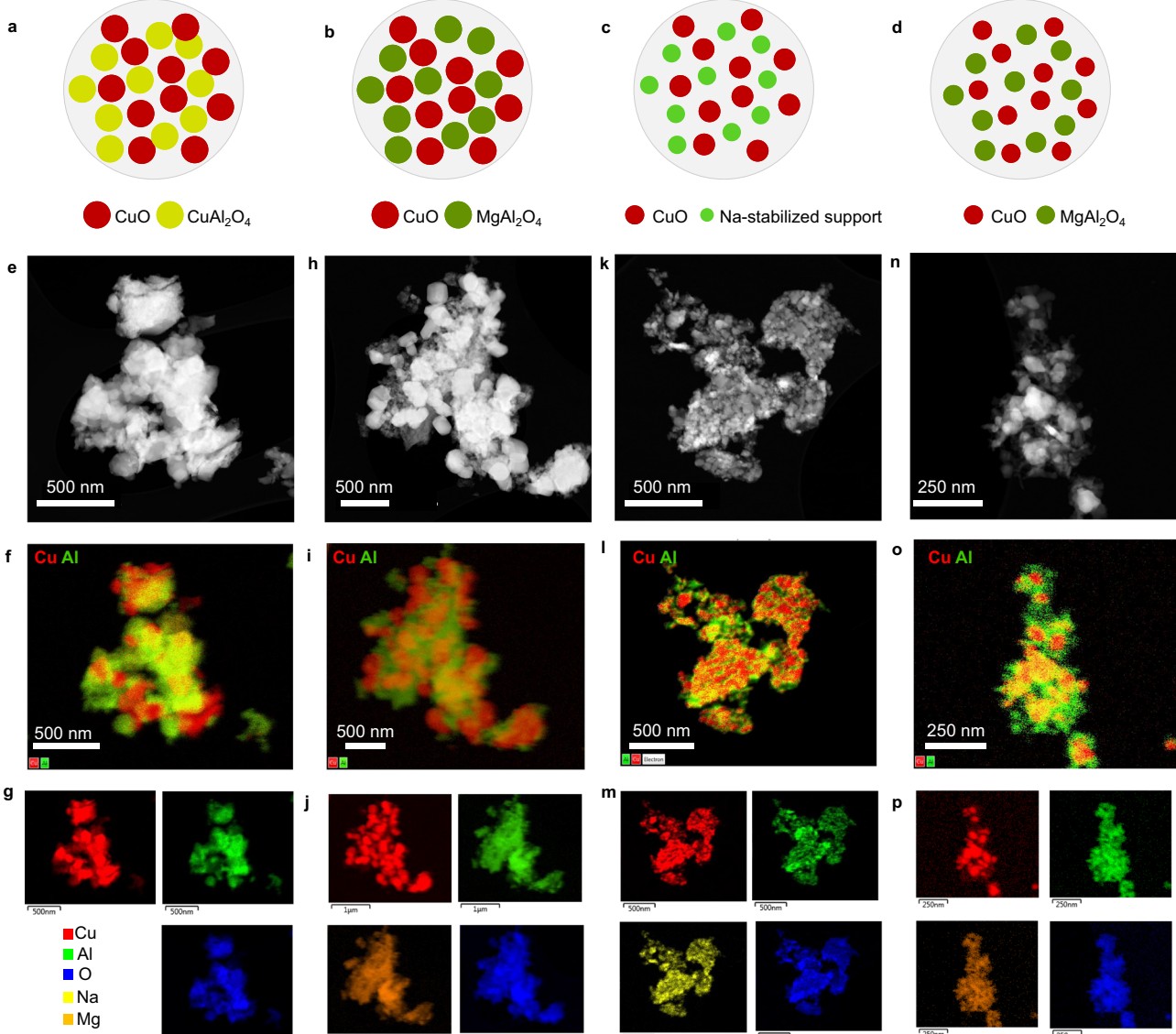

**Fig. 3 | HAADF-STEM imaging and EDS elemental mapping of typical MMOs.**
Diagrams illustrating the composition and phase dispersion of the MMOs: **a** CuO/
$CuAl_2O_4$ derived from CuAl-LDH, **b** CuO/$MgAl_2O_4$ derived from low-crystallinity
CuMgAl-LDH, **c** CuO in sodium-stabilized support, and **d** CuO in $MgAl_2O_4$ support
derived from high-crystallinity CuMgAl-LDH. **e**, HAADF-STEM image and (**f, g**)
element mapping of CuAl-MMO synthesized from CuAl-LDH, consisting of CuO/
$CuAl_2O_4$. Cu:Al=3:2, 2 M NaOH. **h** HAADF-STEM image and **i, j** element mapping of
CuMgAl-MMO derived from low-crystallinity CuMgAl-LDH. Molar ratio of Cu:Mg:Al

is 3:1:2. LDH synthesized with 2 M NaOH reagent. **k** HAADF-STEM image and
**l, m** element mapping of CuAl-MMO derived from CuAl-LDH with residual sodium.
The molar Cu:Al ratio is 3:2 with precipitating agent of 1 M NaOH + 1 M $Na_2CO_3$.
**n** HAADF-STEM image and **o, p** element mapping of CuMgAl-MMO calcined from
high-crystallinity CuMgAl (3:1:2) LDH synthesized with 1 M NaOH and 1 M $Na_2CO_3$.
The combined element maps (**f, i, l, o**) only consist of Cu and Al. All the samples
were calcined at 950 °C for 3 h.

spectra (Supplementary Fig. S3) confirmed the stronger absorbance
associated with interlayer $NO_3^-$ anions when NaOH was used as the
precipitant, while stronger absorbance of interlayer $CO_3^{2-}$ anions was
observed when $Na_2CO_3$ was used (in addition to NaOH) as the pre-
cipitation agent. The varied supersaturation and intercalated anion
compositions have a critical effect on the formation of the LDH pre-
cursors, crystallization, and growth[46,47]. Using 2 M NaOH as the pre-
cipitating agent produced a low-crystallinity LDH (Supplementary
Fig. S2) due to the rapid nucleation and precipitation of ultra-fine low-
crystallinity nanoparticles in the precursor, as observed by SEM
(Supplementary Fig. S9). Whereas high-crystallinity LDH was produced
when using 1 M NaOH + 1 M $Na_2CO_3$ (Supplementary Fig. S2) due to the
low supersaturation and therefore a lower rate of nucleation and
increased growth. The use of $Na_2CO_3$ has also been observed to
increase stacking in the *c*-direction, perpendicular to the LDH sheets,

due to the stronger electrostatic bonds between $CO_3^{2-}$ (*versus* $NO_3^-$)
and the brucite layers. After calcination at high temperatures, the
MMOs prepared using solely NaOH generally showed a lower surface
area compared to those prepared using a mixture of NaOH and $Na_2CO_3$
(Supplementary Fig. S14 and Table S6).

The degree of elemental dispersion in the MMOs critically deter-
mines the redox activity and thermal stability. As visualized in
Fig. 3a−d, Cu mixed oxides derived from different precursors exhibit
varied degree of dispersion. To compare the elemental dispersion and
morphology of the MMOs, we performed HAADF-STEM imaging ana-
lysis (Fig. 3e, h, k, n) and EDS elemental mapping of typical MMOs.
Figure 3 shows typical STEM-EDS images of samples calcined at 950 °C
for 3 h. The CuAl-MMO derived from CuAl-LDH precipitated using
NaOH showed a poorer dispersion of Cu and Al due to the phase
separation of CuO and $CuAl_2O_4$ (Fig. 3f, Supplementary Fig. S10).

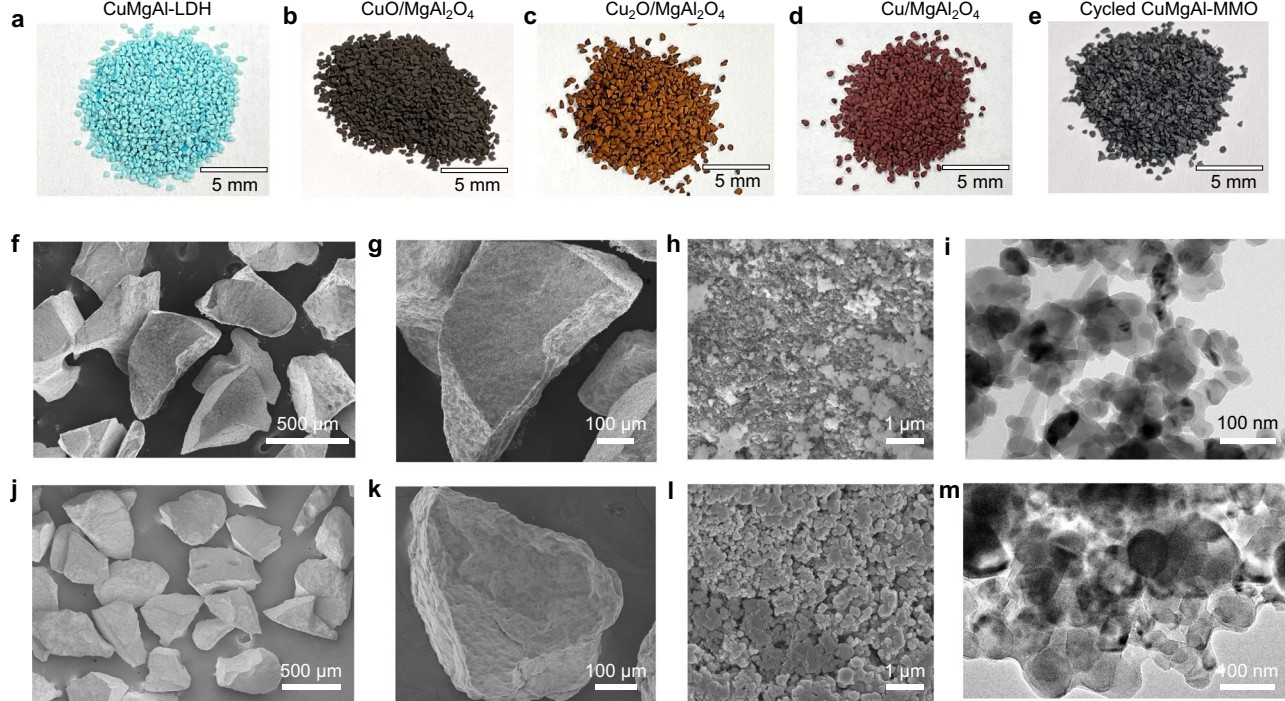

**Fig. 4 | Shape and morphology evolution of CuMgAl-MMO-HS particles.** Photos of CuMgAl-LDH precursors and derived MMOs in different states: **a** LDH precursor powders, **b** freshly calcined particles, **c** decomposed particles (active phase: $Cu_2O$), **d** reduced particles (active phase: Cu) reduction under 2 vol% CO balanced with $N_2$, **e** particles after regeneration (active phase: CuO) after 100 cycles in the FBR. **f, g** SEM image of the freshly calcined particles and **h** their surface, and **i** STEM image. **j–l**,SEM image of the particles at different magnifications, sample recovered after 100 cycles in the FBR, and **m** STEM image of cycled materials.

Similarly, CuMgAl-MMO calcined from low-crystallinity precursor precipitated in NaOH showed a dense aggregation of nanoparticles and the formation of large CuO particles (Fig. 3i, Supplementary Fig. S11). For samples prepared with a precipitating agent containing 1 M NaOH and 1 M $Na_2CO_3$, CuAl-MMO with residual sodium showed the formation of fine CuO nanoparticles dispersed in an amorphous matrix (Fig. 3l, Supplementary Fig. S12). In contrast, CuMgAl-MMO calcined from high-crystallinity CuMgAl-LDH showed homogeneous distribution of CuO (about 30–40 nm) at the nanometer scale (Fig. 3o, Supplementary Fig. S13). A higher degree of dispersion helps to stabilize the CuO phase and improves the resistance to sintering, and hence allows for a higher redox activity and faster kinetics.

**Optimization of mechanical properties**
To use the LDH-derived MMOs as oxygen storage materials in FBRs, we further optimized the mechanical strength of the materials. A new batch of CuMgAl-LDH (Cu:Mg:Al molar ratio of 3:1:2) was synthesized at a higher pH value (pH of 11.0) during the co-precipitation (Fig. 4a). XRD analysis and SEM imaging confirmed the LDH structure and nanoplatelet morphology (Supplementary Fig. S15). Though, the platelets were more densely aggregated compared to those of the precursors prepared at the lower pH of 9.5, as also indicated by narrower pore size distribution derived from $N_2$ adsorption (Supplementary Fig. S15). Figure 4b shows the calcined MMOs after crushing and sieving into smaller particle size ranges. XRD analysis confirmed that the calcined MMOs consisted of CuO and $MgAl_2O_4$. The decomposed particles had a homogeneous orange colour (Fig. 4c), while the fully reduced samples were brownish red (Fig. 4d), being distinctive for $Cu_2O$ and Cu, respectively. An increase of the pH value set during precipitation from $9.5 \pm 0.2$ to $11.0 \pm 0.2$ resulted in the formation of smaller LDH crystals, which further aggregated into dense particles after calcination as observed by SEM (Fig. 4f–h) and STEM imaging (Fig. 4i). Furthermore, the crushing strength of the calcined particles increased from $2.3 \pm 1.1$ N to $5.7 \pm 2.6$ N. Henceforth, the non-optimized

lower strength (LS) sample (prepared at *pH* of 9.5) and the optimized higher strength (HS) sample (prepared at pH of 11.0) will be referred to as CuMgAl-MMO-LS and CuMgAl-MMO-HS, respectively. The particles with the higher mechanical strength are more suitable for operation in FBRs, where higher mechanical forces occur. As shown in Fig. 4e, the cycled particles maintained the initial shapes and sizes over 100 redox-cycles. SEM imaging of the cycled samples (Fig. 4j–l) suggested that large grains were formed due to sintering at high temperatures, yet the STEM analysis showed that the CuO nanoparticles were still well dispersed in the support, as shown in Fig. 4m.

**Cyclical reduction and oxidation of the oxygen sorbents**
To investigate the reactivity and phase changes of the CuMgAl-MMOs as redox sorbents, isothermal reduction and oxidation reactions were performed in a TGA and an FBR (their set-ups are illustrated in Supplementary Figs. S16 and 17, respectively). The phase changes during the reduction and oxidation were tracked by ex situ XRD. Figure 5a shows the weight change of oxygen sorbents over 100 redox cycles between CuO and $Cu_2O$, which were performed by decomposition in $N_2$ and oxidation in air for a total of >1000 min (further TGA profiles are shown in Supplementary Fig. S18). We further investigated the stability of the MMOs during redox-cycling with deep reduction of CuO to Cu under 5 vol% $CO/N_2$, followed by oxidation in air. The oxygen storage capacity of each sample tested under full reduction and re-oxidation (CuO↔Cu) remained highly stability over the 100 cycles (*e.g.* 13 wt% for CuMgAl-MMO; Supplementary Fig. S18). A typical $O_2$ release and storage profile of MMOs in a TGA is shown in Fig. 5b. Both CuMgAl-MMO samples (regardless of their mechanical strength) showed faster rates of decomposition and oxidation than those observed for the CuAl-MMOs (consisting of CuO and $CuAl_2O_4$). The slightly slower rate of decomposition and oxidation for CuMgAl-MMO-HS, in comparison to CuMgAl-MMO-LS, is likely a result of the slightly lower porosity and denser aggregation of nanoparticles. Unlike CuAl-MMO, which deactivated very quickly during redox-cycling (Fig. 5c), the CuMgAl-MMO

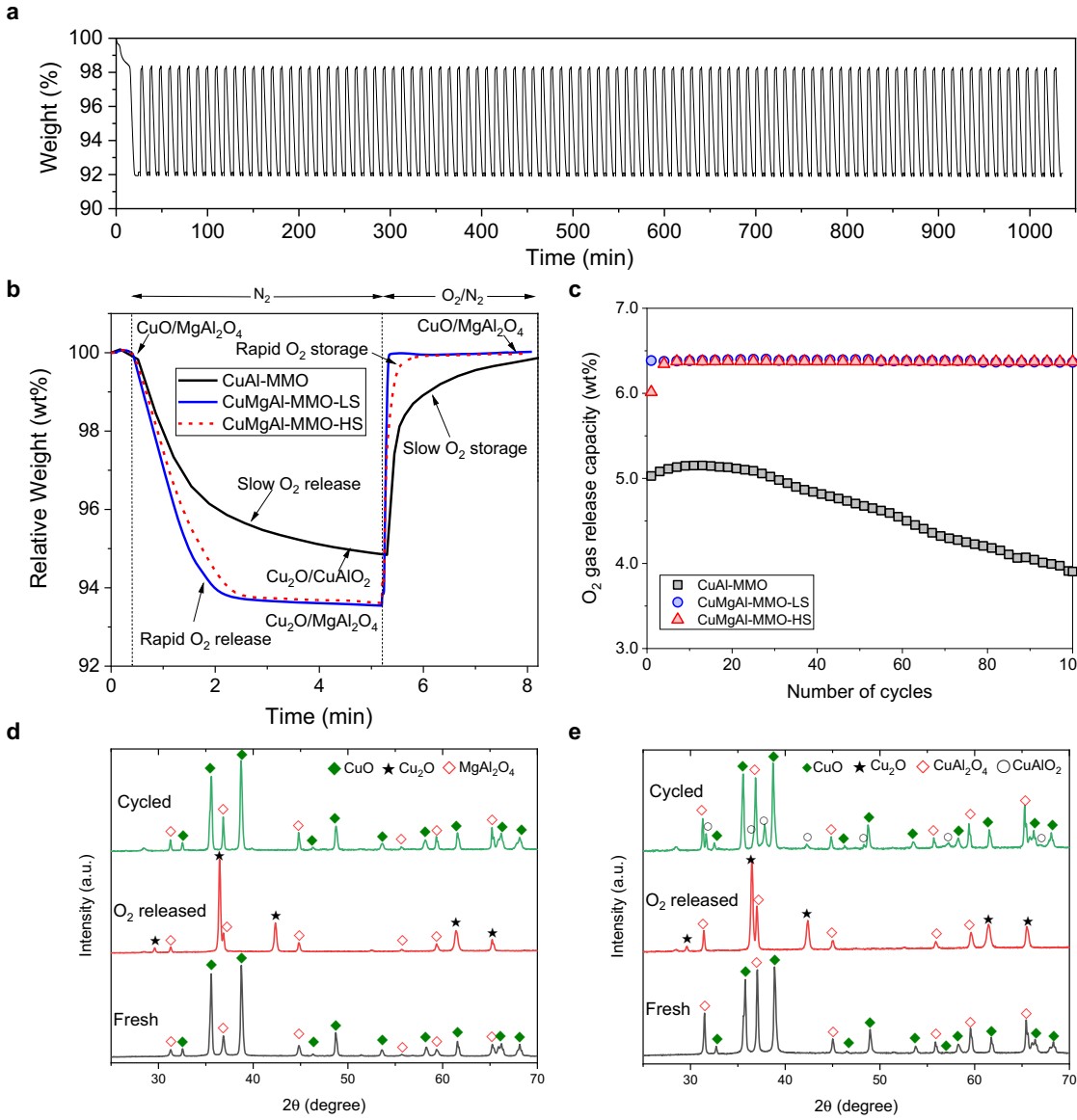

**Fig. 5 | Redox cycling of oxygen storage materials in a TGA. a** Weight change over 100 cycles of $O_2$ release (CuO to $Cu_2O$) under $N_2$ of CuMgAl-MMO-LS. **b** weight change profile during $O_2$ release under $N_2$. Three typical samples were tested: CuAl-MMO, CuMgAl-MMO-LS (low strength), and CuMgAl-MMO-HS (high strength). The tests were performed in the TGA at 900 °C. **c** $O_2$ release capacity over 100 cycles of $O_2$ release. **d** XRD patterns of CuMgAl-MMO-LS at different states; **e** CuAl-MMO in different redox states.

samples showed almost no deactivation over 100 continuous redox cycles, i.e. neither a significant reduction in observed rates nor in the observed $O_2$ release capacity. The $O_2$ release capacities of the CuMgAl-MMO-LS and CuMgAl-MMO-HS remained constant at 6.4 wt% over 100 redox cycles. The decline of the $O_2$ release capacity of the binary MMO can mainly be attributed to the formation of $CuAl_2O_4/CuAlO_2$, which have significantly slower rates of decomposition and re-oxidation than $CuO/Cu_2O$[48].

XRD analyses of the samples confirmed the chemical stability of $CuO/MgAl_2O_4$ in the CuMgAl-MMO during redox cycling (Fig. 5d, Supplementary Figs. S19 and S20). Owing to the stabilization by the $MgAl_2O_4$ support, CuO decomposed to $Cu_2O$ and was regenerated to CuO without forming $CuAlO_2$. In contrast, formation of $CuAl_2O_4$ in the CuAl-MMO was verified by XRD (Fig. 5e, Supplementary Fig. S20). The $O_2$ release and storage reaction mechanisms of the CuAl-MMO are discussed in the supplementary information (Supplementary Fig. S21). The phase changes were also studied by in situ high-temperature XRD by exposing the metal oxides to dynamically switching $N_2$ and air gas

streams in the XRD stage (Supplementary Fig. S22). The results confirmed the reversible and fast $O_2$ release and storage in CuMgAl-MMO and relatively slow kinetics of CuAl-MMO due to formation of $CuAl_2O_4$. The morphology evolution of the MMO was studied by SEM of the cycled samples (Supplementary Figs. S23 and S24). Generally, all MMOs show a certain degree of sintering as aggregated grains were observed on the surface of the particles, while the nanoplatelet-like morphology derived from LDH precursors could be preserved in local regions.

## Reversible and stable $O_2$ release and storage in a fluidized bed reactor

To assess the chemical and thermal stability of the oxygen sorbents, three typical samples (CuAl-MMO, CuMgAl-MMO-LS, and CuMgAl-MMO-HS) were tested for 100 cycles of alternating decomposition and oxidation in a FBR at 900 °C (Fig. 6, Supplementary Fig. S25), with one cycle of full decomposition and oxidation every 20 cycles. Due to the excessive attrition of the CuAl-MMO particles, operation

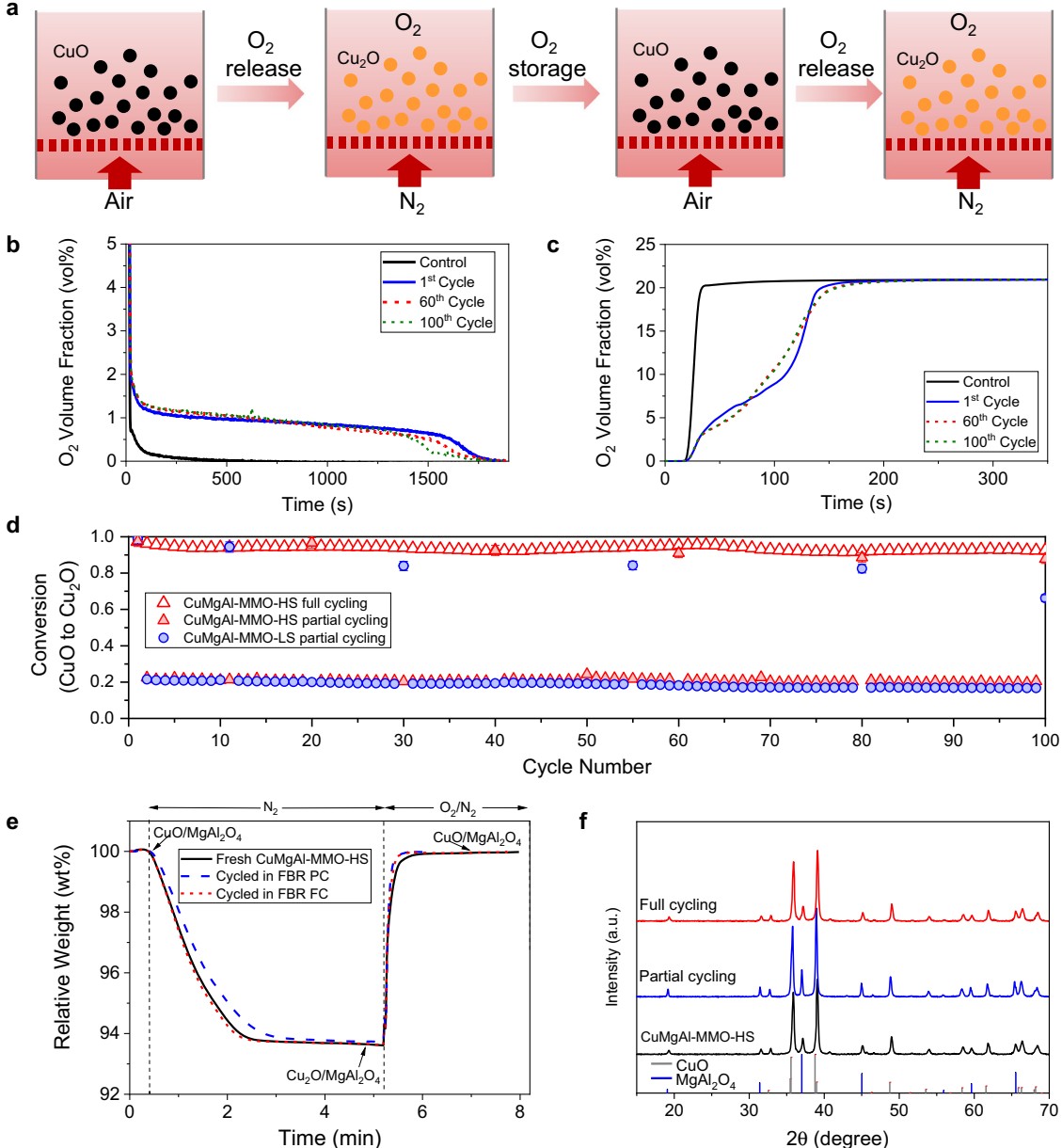

**Fig. 6 | Redox cycling of oxygen storage materials in an FBR.** Two typical samples were tested: CuMgAl-MMO-LS (low strength), and CuMgAl-MMO-HS (high strength). **a** Schematic diagram showing the cyclic $O_2$ release and storage due to phase changes of CuO and $Cu_2O$ in the MMOs. **b** $O_2$ volume fraction of gas during reduction and **c** oxidation of oxygen storage materials in the FBR. Control experiments were performed with quartz sand loaded in the reactor without the MMO. **d** Conversion of CuO to $Cu_2O$ over 100 redox cycles in the FBR, where partial cycling was performed by 100 cycles of partial decomposition and oxidation with periodical full decomposition of CuO every 20 cycles, and full cycling was performed by full decomposition and oxidation for 100 cycles. **e** TGA profiles of freshly calcined particles, particles cycled for 100 cycles in a FBR with partial decomposition (FBR PC) and oxidation with full decomposition every 20th cycle in an FBR; and particles cycled for 100 cycles with full decomposition and oxidation in an FBR (FBR FC). **f** XRD patterns of fresh and cycled CuMgAl-MMO-HS particles.

was only possible for 9 cycles. Over 100 cycles, the overall conversion of the full reduction cycles for the CuMgAl-MMO decreased for the low-strength and high-strength samples from 0.98 to 0.62 and 0.97 to 0.82, respectively. The observed attrition of the low-strength sample agreed well with a higher porosity and a lower crushing strength. Additionally, the high-strength sample was rigorously tested in a FBR for 100 full decomposition cycles (i.e. for *ca.* 65 h). As shown in Fig. 6d, the conversion remained constant at $0.90 \pm 0.05$ (of the theoretical conversion) without significant attrition or agglomeration occurring. In another experiment, the high-strength sample CuMgAl-MMO-HS was tested over 500 cycles in the TGA (Supplementary Fig. S26). The near-constant conversion confirmed the stability of the materials over extended redox-cycling and highlighted

the potential of CuMgAl-MMOs as promising candidates for further development and scale-up.

To understand the structural evolution during redox cycling, we collected CuMgAl-MMO particles cycled in the FBR and investigated their structural changes. The slight decrease in conversion observed in the FBR may be attributed to three main factors: (i) loss of material when unloading the bed, (ii) attrition suffered by the oxygen carriers, (iii) loss of active material due to the formation of $CuAl_2O_4$. The CuMgAl-MMOs sampled after FBR experiments were subjected to further TGA cycling to measure the $O_2$ release capacity (Fig. 6e), which remained at 6.4 wt%. This, and the absence of CuAl spinel oxide peaks during post-cycling XRD analysis (Fig. 6f) confirmed that formation of these spinel phases (for the CuMgAl-MMOs) during redox-cycling in

the FBR did not occur. The ratio of active and support phases calculated for fresh and cycled CuMgAl-MMOs using a Rietveld refinement also remained unchanged (Supplementary Table S7). SEM images of the particles and particle surfaces before and after cycling are shown in Fig. 3. The surface morphology of the samples recovered after redox cycling is slightly smoother than the surface of the freshly calcined material, which indicated slight occurrence of surface attrition in the FBR. After 100 cycles of partial decomposition and oxidation in the FBR, the evolved surface morphology showed the aggregation of platelets into grains (Fig. 3l). The STEM images of the redox cycled CuMgAl-MMO-HS particles confirmed the nanoscale dispersion of the CuO and $MgAl_2O_4$ spinel phases (Fig. 3m). Mercury intrusion porosimetry (MIP; Supplementary Fig. S27) and helium pycnometry tests suggested that the porosity of the CuMgAl-MMO-LS and CuMgAl-MMO-HS samples decreased from 0.74 to 0.64, and 0.38 to 0.34, respectively. $N_2$ adsorption and MIP also indicated the changes of pore size distributions for each material, showing the collapse of mesoporous pores and a shift towards larger pore diameters. The results of this study confirmed the high chemical stability and high thermal stability of the CuMgAl-MMO-HS during redox cycling although the mechanical properties of the materials need to be further improved by particle engineering and manufacturing in the future.

## Chemical looping combustive purification

To further demonstrate the practical application of these nanostructured MMOs as oxygen sorbents, we performed cyclic reduction and oxidation of the MMOs using low concentrations of CO and $CH_4$ (Fig. 7a), which is relevant to many important industrial processes. In this work, we tested the materials during redox cycling with low concentrations of CO and $CH_4$ balanced with $N_2$ at low temperatures (400 °C) and high temperatures (800 °C) in the FBR. The typical gas concentrations of the reactive gases during the reduction and oxidation stages are shown in Fig. 7b–d (additional gas profiles are presented in Supplementary Figs. S28, S29). As shown in Fig. 7b, the CuMgAl-MMO can efficiently oxidize low concentrations of CO to $CO_2$ at 400 °C. The residual CO/$CH_4$ concentrations are usually below the detection limit. The CO and $CH_4$ conversions at 400 °C and 800 °C were close to unity (Fig. 7e). To avoid coke deposition during the reduction of $CH_4$, the degree of conversion of CuO was controlled at about 60%. The materials also demonstrated high stability for 20 cycles over 1000 min with no agglomeration of the oxygen carriers being observed in the samples withdrawn from the reactor. XRD patterns of the materials cycled with $CH_4$ and CO (Supplementary Fig. S30) also found no peaks matching with Cu-Al oxides ($CuAl_2O_4$ or $CuAlO_2$), demonstrating the phase stability at CLCP conditions. These results confirmed the high reactivity and stability of the Cu-based redox sorbents, which have shown great potential for application in industrially important processes such as argon recovery and recycling in silicon manufacturing processes (Fig. 7f)[26].

## Discussion

The structures and properties of precursors crucially determine the performance of redox sorbents in thermochemical looping processes. We have demonstrated that by rationally tuning the chemistry of LDH precursors and derived MMOs, redox sorbents with a high reactivity and excellent stability in thermochemical redox cycles can be synthesised. Tuning the chemistry of the CuMgAl-LDH precursors resulted in the formation of highly dispersed CuO phases in a stable Mg-Al spinel oxide support, which effectively inhibited sintering, agglomeration and the undesired formation of $CuAl_2O_4$ over extended redox-cycling periods. The CuMgAl-MMOs showed stable $O_2$ release and oxygen storage capacities. The results from long-term redox-cycling experiments at a high temperature of 900 °C in the TGA and the FBR suggested that CuMgAl-LDH-derived MMOs are promising oxygen

storage materials for various applications. The efficient and stable redox-cycling over a wide temperature range (400–800 °C) and at low combustible gas concentrations demonstrated the potential of these MMOs as redox sorbents for gas purification processes, which may have important implications on the design of future industrial gas purifications systems over a wide range of energy processes.

Despite extensive TEM and EDS analysis, it remains challenging to fully understand the interfaces between the Cu phases and the support due to difficulties of characterizing the sintering behaviour at high temperatures. In this work, the CuO nanoparticles were well dispersed in the support, such embedding effect were observed for CuO/$ZnAl_2O_4$ catalysts derived from CuZnAl hydrotalcites[49]. According to thermogravimetric analysis (Figs. 5b and 6e) as well as ex situ and in situ XRD analysis (Fig. 5d, and Supplementary Figs. S19, S20, S22), the gaseous oxygen is predominantly released from crystalline CuO, confirming that the content of $CuAl_2O_4$ or $Mg(Cu)Al_2O_4$ is negligible. Nevertheless, residual Cu species may be incorporated into the spinel support at the grain boundaries of the CuO nanoparticles and Mg-Al spinel support, which likely form reducible $CuAl_2O_4$ or $Mg(Cu)Al_2O_4$ overlayers, which in turn function as physical barrier layers to separate CuO nanoparticles from sintering or further immobilise Cu phases in the support. These strong active phase-support interactions and the local electronic transfer merit further study in the future.

In this work, we have mainly focused on the fundamental understanding of the precursor chemistry and its relationship with structures and properties of the MMOs in redox cycles. The mixed oxides were prepared by calcination of the precursors and subsequent crushing and sieving to obtain suitable particle size ranges for use in FBRs. Although the Cu-based particles maintained their physical structures during our laboratory-scale testing, we have to consider the mechanical properties and shapes of particles used in different types of reactors. For chemical looping processes using FBRs, attrition of particles is one of the most critical obstacles that limit commercialization. The redox reactions at high temperatures generate significant chemical stresses while thermal sintering generates additional mechanical and thermal stresses, which tend to induce structural changes and lead to a decline of the mechanical strength of the redox sorbents in reactors, such as particle breakage and attrition. Advanced particle manufacturing technologies (that enable production of near-spherical particles), such as spray drying and granulation, might further improve the mechanical resistance and reduce attrition. For use in packed bed reactors, large-scale pelletizing techniques could be used to prepare suitable particle shapes with enhanced mechanical strength.

Although our work focused on the applications of LDH-derived MMOs in thermochemical processes, the fundamental understandings of the precursor chemistry reported in this work have broad implications on other applications of LDHs and LDH-derived materials. In recent years, transition metal-based LDH materials have shown great promise as (photo-)electrocatalysts (or their precursors)[50,51]. For example, one recent work reported CuAl-LDH-derived copper electrocatalysts that demonstrated promising acetylene electroreduction performance[52]. The CuAl-LDH was prepared using NaOH and $Na_2CO_3$ as precipitating agents, which would very likely lead to formation of sodium-containing species (i.e. dawsonite, $NaAl(CO_3)(OH)_2$) according to the findings of this work and our previous work[40,53]. However, the effect of the residual sodium impurities on the structure of electrocatalysts and electrochemical reactions was poorly understood. Our work shows that the fundamental understanding of precursor chemistry and the relationship with compositions and structures of MMOs could allow for a rational design of the materials, leading to significantly improved reactivity and stability.

In summary, this work demonstrates the precursor engineering of nanostructured oxygen storage materials with significantly

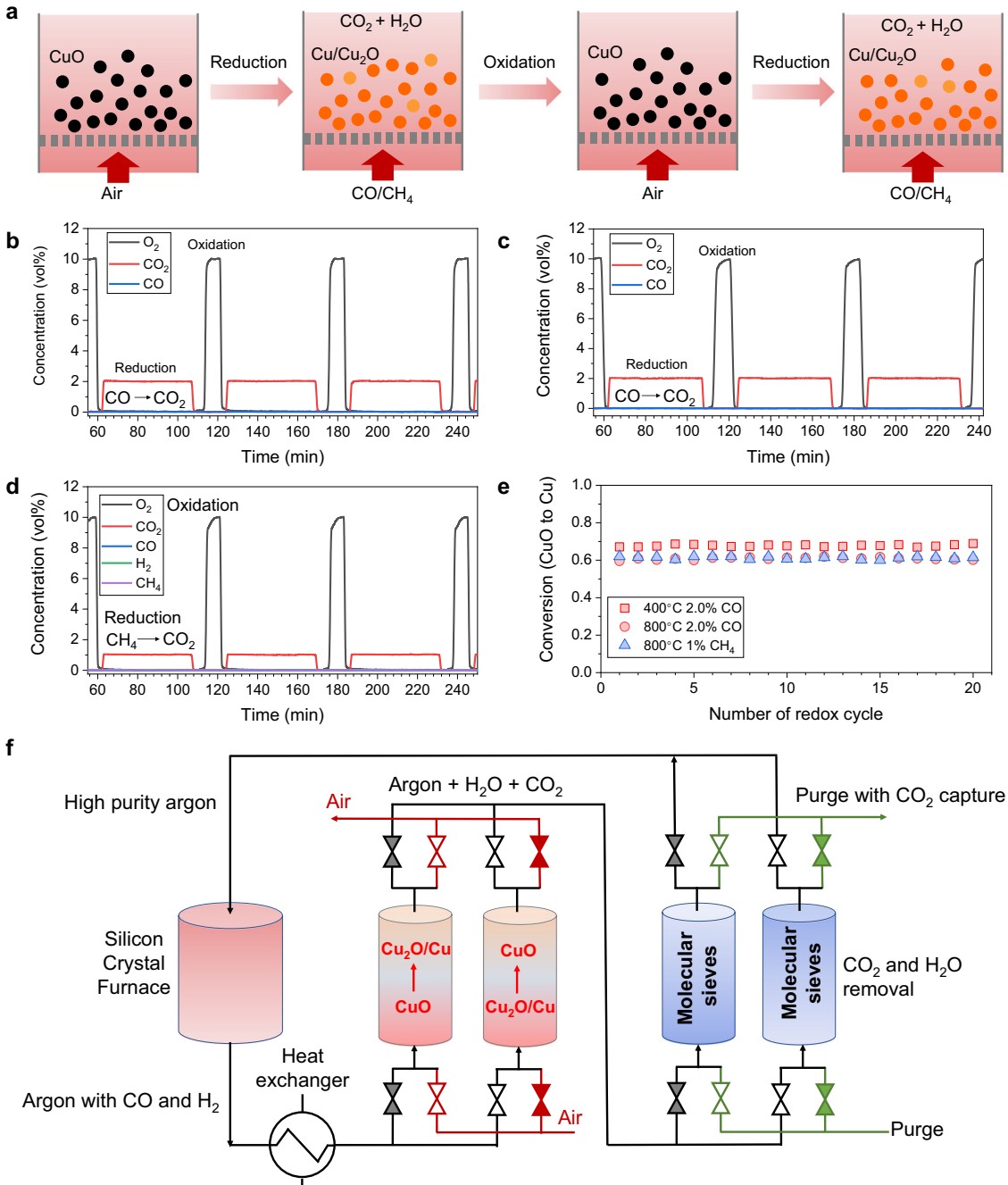

**Fig. 7 | Efficient gas purification by thermochemical redox cycling of MMOs.**
**a** Schematic diagram of the main redox-reactions of MMO upon reduction in
diluted CO and CH$_4$, and oxidation under air, **b–d** Gas concentration profiles during
redox-experiments with reduction in 2.0 vol% CO at (b) 400 °C and (c) 800 °C; and
**d** reduction in CH$_4$ at 800 °C. **e** Conversion of oxygen sorbents *versus* number of
redox cycles. **f** Schematic diagram of CLCP for argon recovery from silicon man-
ufacturing processes, consisting of two stages (1) removal of contaminated gases
by copper-based oxygen storage materials and (2) removal of CO$_2$ and H$_2$O with
pressure swing adsorption.

enhanced cycling stability for thermochemical looping applica-
tions. The structural diversity of LDHs allows versatile combination
of metal ions and further incorporation of high dispersion of cat-
alytic active metal species for development of redox sorbents
and catalysts. We expect that the synthesis strategy of this
work may inspire the production of highly stable oxygen storage
materials, sorbents, and nano-catalysts for many emerging
energy processes based on thermochemical looping processes,
including chemical looping reforming of methane, CO$_2$ utilisation,
hydrogen production, thermochemical energy storage, and redox
catalysis.

## Methods
### Synthesis of materials
The LDH precursors were synthesised by co-precipitation at
constant pH. A metal nitrate solution with a total metal ion con-
centration of 2 M was prepared by dissolving Cu(NO$_3$)$_2$·2.5H$_2$O,
Al(NO$_3$)$_3$·9 H$_2$O, and Mg(NO$_3$)$_2$·6 H$_2$O in the desired ratios in de-
ionised water (DI water). An aqueous alkaline solution was pre-
pared that contained 1 M NaOH and 1 M Na$_2$CO$_3$. A series of
control samples were also prepared using an aqueous 2 M NaOH
solution as the precipitating agent. The metal nitrate solution and
alkaline solution were mixed by dropwise addition of both

solutions into a reaction beaker under constant stirring (at 300 rpm). A pH value of $9.5 \pm 0.2$ was maintained by adjusting the flow rate of the acidic and alkaline solutions. The pH value was measured using a calibrated Orion Star A211 (Thermo Scientific) system with its probe being submerged in the product mixture. After complete addition of the metal nitrate solution, the mixing was stopped, and the suspension was aged for 3 h at room temperature. Then, the supernatant was carefully decanted off and the precipitate was mixed with fresh DI water. This washing procedure was repeated until a conductivity below 50 $\mu S\,cm^{-1}$ was measured. The precipitated solids were separated from the liquid phase first by decanting and then by vacuum filtration. The solids were then dried at 60 °C for at least 12 h in a ventilated oven. After drying, the samples were calcined in a horizontal tube furnace at 950 °C for 3 h in an airflow of about $1\,L\,min^{-1}$ at standard atmospheric temperature and pressure (SATP, i.e. 20 °C and 101.3 kPa), using a heating rate of $15\,°C\,min^{-1}$. After calcination, the solids were crushed and sieved into the desired particle size ranges. For some samples, the calcination temperature was varied (400, 600, 800 °C) to study the evolution of phases and morphologies.

## Characterisation techniques

$N_2$ adsorption analysis was carried out in a Micromeritics ASAP 2000 and the data were used to calculate the specific surface area using the Brunauer–Emmett–Teller method[54]. The samples were first degassed at 110 °C under vacuum for 12 h, and then further degassed at 110 °C for another 12 h after being loaded into the apparatus. Nitrogen adsorption isotherms were then measured at 77 K. FTIR was performed using the KBr pellet technique. The surface morphology of materials was studied using a scanning electron microscope (SEM) type LEO Gemini 1525 or Hitachi S5500. Prior to SEM analysis, the samples were sputter-coated with a thin layer of gold (for 30 s at 20 mA). The morphology and chemistry of the particles was analysed by using a JEOL JEM-2100F TEM operated at 200 kV. Bright Field (BF), High-Angle Angular Dark Field (HAADF) and EDS maps were obtained for different samples. Powder XRD analysis was conducted using an X'Pert PRO (PANalytical) with Cu-Kα radiation ($\lambda = 0.1541$ nm). The crystallite size was estimated using Scherrer's equation ($L = K/(B \cos \theta)$)[55], and the peak broadening, $B$, used in the calculation was taken as the difference of the observed peak broadening and the instrument broadening ($2\theta = 0.102°$). The latter was determined experimentally using the (111) peak of a sapphire standard. The $d$-spacing, $d$, was calculated from the XRD data using Bragg's law ($\lambda = 2d (\sin \theta)$). In-situ XRD measurements were carried out using an Empyrean (PANalytical) under Bragg-Brentano orientation with Cu-Kα radiation and within a hermitically sealed Anton-Paar HTK 1200N furnace chamber. Gas flow was controlled at $50\,ml\,min^{-1}$ by two digital mass flow controllers and the gas switching was controlled by the FlowView program (Bronkhorst). Redox kinetics were investigated by repeatedly scanning the $2\theta$ range; 34 to 40° at a step size of 0.013°. Each scan had a duration of approximately 90 s once the goniometer had repositioned to the start angle. Elemental analyses were carried out using ICP-MS (Agilent 7900) calibrated with standards for Cu, Mg, Al, and Na. The samples were prepared in highly sterile single-use sample flasks by dilution with a mixture of ultra-pure water and nitric acid (2 vol%). XRF was conducted using an AXS S4 Explorer (Bruker). The results from the XRF analysis were converted from mass fractions of the elements to mass fractions of the oxides using the relative molecular weights of the compounds.

## Thermogravimetric analysis

Thermogravimetric analysis was carried out in a Q5000 (TA Instruments) to evaluate the chemical stability of the metal oxides (Supplementary Fig. S16). In a typical experiment, 3–4 mg of the calcined mixed oxides were placed in a platinum crucible (1.5 mm

high and 9.8 mm in diameter) and cycled for 100 redox cycles. For CLOU experiments, the decomposition was carried out under $N_2$ for 5 min, followed by re-oxidation in air for 3 min. For CLC experiments, the metal oxides were reduced for 7 min in 5 vol% CO balanced with $N_2$. This step was followed by a 1 min $N_2$ purge. The samples were then re-oxidised for 1 min in air. After a further purge (1 min) in $N_2$, the four-step sequence was repeated for a total of 100 cycles. All experiments were performed at 900 °C, and the total gas flow rate was kept constant at $200\,ml\,min^{-1}$ (SATP). The observed oxygen release capacity (CLOU) and oxygen storage capacity (CLC) were defined as the difference of the relative weights at the beginning and the end of the decomposition and reduction period, respectively.

## $O_2$ release and storage in a fluidised bed reactor

In addition to experiments in the TGA, a cylindrical quartz FBR with internal diameter of 30 mm was used to examine the long-term cycling performances of the oxygen storage materials[17]. The reactor system is illustrated in Supplementary Fig. S17 and has been described elsewhere[56]. In addition to thermo-chemical stresses due to the chemical reaction, the MMOs particles in the FBR are also exposed to minor mechanical stresses owing to collisions of particles with other particles and/or the reactor walls. In a typical FBR experiment, 15 g of MMOs in the particle size range 300–425 μm were used. First, inert cycles were carried out by repeated automated gas switching between $N_2$ and air in the absence of the MMO particles. Then, the metal oxides were added to the reactor and cycled for 100 redox cycles at 900 °C. Two cycling modes were used to test the stability of the metal oxides: partial cycling and full cycling. Partial cycling comprised of five, 20-cycle segments consisting of 19 short cycles where the decomposition period (360 s) was not long enough to fully decompose the mixed oxides, followed by one longer cycle where the decomposition period (1800 s) allowed complete decomposition of the oxygen carriers. The full cycling mode consisted of 100 cycles of decomposition and oxidation where the time was sufficient for complete decomposition of the oxygen carriers during each cycle. The molar fraction of $O_2$ calculated from the gas analyser measurements of the gas leaving the reactor during inert cycles ($y_{O_2,inert}$) and during redox cycling of the oxygen storage materials ($y_{O_2,sample}$) were used to calculate the reaction rates by,

$$X_{O_2,release} = \frac{MW(O_2)}{m_{ox} R_{O_2}} \int_0^t n_{out}(y_{O_2,sample} - y_{O_2,inert}) dt \qquad (1)$$

where $m_{ox}$ is the mass of oxygen storage materials loaded into the reactor (g); $n_{out}$ is the molar flow rate of the effluent gas in $mol\,s^{-1}$; MW($O_2$) is the relative molecular mass of $O_2$ (32 g mol$^{-1}$) and $R_{O_2}$ is the gaseous $O_2$ release capacity determined by thermogravimetric analysis (wt%).

## Reaction with gaseous fuels and $O_2$ storage in a fluidised bed reactor

The cycling performances of the oxygen storage materials under complete reduction conditions were analysed in an FBR. Two gaseous fuels were investigated, 1 vol% $CH_4/N_2$ at 800 °C, and 2 vol% $CO/N_2$ at 400 and 800 °C. In a typical FBR experiment, 15 g of MMOs in the particle size range 300–425 μm were used. Inert cycles were carried out by repeated automated gas switching between gaseous fuel, $N_2$ purge, air, and another $N_2$ purge without the presence of the samples. Then, the metal oxides were added to the reactor and cycled for 20 redox cycles of full reduction to Cu and re-oxidation to CuO. For CLCP cycling, one cycle consisted of a reduction period of 45 min to limit coke deposition, followed by an $N_2$ purge (2.5 min) to prevent mixing of $O_2$ and the reducing gas, oxidation in 10 vol% $O_2$ (12 min), and another $N_2$ purge (2.5 min). The total gas flow rates were $2,250\,ml\,min^{-1}$

and 2,000 ml min⁻¹ at SATP for the CLCP experiments using CO at 400 and 800 °C, respectively, and 1000 ml min⁻¹ for CH₄ at 800 °C. The conversion of metal oxides (CuO to Cu) during reduction period was calculated by the ratio of total amount of oxygen reacted with gaseous fuel to the total amount of available oxygen in the oxygen storage materials. The conversion of oxygen storage materials for CO ($X_{\text{red,CO}}$) and CH₄ reduction ($X_{\text{red,CH}_4}$) are:

$$X_{\text{red,CO}} = \frac{MW(O_2)}{2m_{\text{ox}}R_O} \int_0^t n_{\text{out}} y_{CO_2,out} dt \qquad (2)$$

$$X_{\text{red,CH}_4} = \frac{2MW(O_2)}{m_{\text{ox}}R_O} \int_0^t n_{\text{out}} y_{CO_2,out} dt \qquad (3)$$

where $m_{\text{ox}}$ is the mass of oxygen storage materials loaded into the reactor (g); $n_{\text{out}}$ is the molar flow rate of the effluent gas in mol s⁻¹; $MW(O_2)$ is the relative molecular mass of $O_2$ (32 g mol⁻¹) and $R_O$ is the oxygen storage capacity determined by thermogravimetric analysis (wt%). $y_{CO_2,out}$ is the mole fraction of $CO_2$ in the product gas.

## Data availability
The authors declare that all the data supporting the findings of this study are available within the paper and its Supplementary Information files or from the corresponding author upon request.

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

## Acknowledgements

This work was funded by the EPSRC Programme EP/P026214/1, EP/W002841/1, EP/V047078/1, and EP/T033940/1, and EPSRC studentship EP/R513052/1. This project also received funding from the European Research Council (ERC) under the European Union's Horizon 2020 research and innovation program (grant agreement No 851272, ERC-StG-PE8-NanoMMES). The authors thank the Department of Chemical Engineering at Imperial College London for the funding of a Ph.D. scholarship in support of this project. M.H. acknowledges the support of one EPSRC DTP scholarship. G.E.W. acknowledges the support of EPSRC Centre for Doctoral Training (CDT)—Fuel Cells & Their Fuels. Q.S. acknowledges the support of Imperial College Research Fellowship and Department of Chemical Engineering start-up funding. D.Z. acknowledges the support of the China Scholarship Council, National Natural Science Foundation of China (Grant no. 51906041), and the Natural Science Foundation of Jiangsu province (Grant no. BK20190360). R.X. acknowledges the National Science Foundation for Distinguished Young Scholars of China (Grant no. 51525601).

## Author contributions

Q.S., P.S.F., and R.X. conceived and supervised the research. M.H. performed synthesis of materials and carried out most chemical looping experiments. C.F.P. performed kinetic experiments, materials analysis and characterization. L.Z. performed synthesis and characterization. N.D., K.H.H.C., and Z. Z. contributed to materials synthesis and testing. D.Z. helped with the materials synthesis and characterization under the supervision of R.X. and Q.S. O.G.D. contributed to STEM, HR-TEM, and EDX analysis. G.E.W., A.V.B. and S.J.S. contributed to in situ high temperature XRD measurements. K.L.S.C. discussed and analysed the results as well as commented on the manuscript. Q.S. performed most SEM analyses. Q.S., M.H., C.F.P., and P.S.F. wrote the manuscript with contribution from co-authors. All authors participated in manuscript preparation, data interpretation, and discussions.

## Competing interests

The authors declare no competing interests.
