## [Peer Review File · Nature Communications]

Title: Precursor engineering of hydrotalcite-derived redox sorbents for reversible and stable thermochemical oxygen storageREVIEWER COMMENTS

Reviewer #1 (Remarks to the Author):

Comments on Nature Communications manuscript NCOMMS-21-47225

This manuscript describes the synthesis and precursor engineering of hydrotalcite-derived copper-based nanostructures for reversible and stable oxygen storage in chemical looping processes for combustion and gas purification. This research provides a comprehensive approach, combining characterization using SEM, STEM, BET, FTIR and XRD with performance test using TGA and fluidized bed reactor. However, the novelty of this manuscript is relatively low considering the CuO/MgAl₂O₄ oxygen carrier has been extensively applied to chemical looping combustion in the context of the oxygen uncoupling scheme for combustion (recent examples: Kuang et al, Energy & Fuels, 35(1): 618-625, 2020; Zheng et al, Energy & Fuels, 34(7): 8718-8725, 2020). Additionally, the nanostructure in precursors, which was claimed as a key feature of this work, is very common in coprecipitation synthesis of materials. The authors were unable to reveal the reaction chemistry of or the mechanism underlining how Na and Mg ions inhibit CuAl₂O₄ formation, which is the key to and the most interesting part of this work. As such, this work is premature and not recommended for publication in a premier journal such as Nature Communication. Some other specific comments are given below:

1. The nanostructures in the precursors were not investigated properly. The STEM images in Fig 2 show amorphous structure with no obvious lattice fringes, which is contradictory to the XRD observation that the oxygen carrier is highly crystalline.
2. This work was flooded with SEM images, which is very unnecessary. No detailed nanostructure information was provided.
3. The redox cycles in this work has been tested up to 100 cycles in TGA and 20 cycles in FBR. This is significantly less compared with the state-of-the-art chemical looping systems that use CuO/MgAl₂O₄ as oxygen carriers, such as Zheng, et al, Journal of Energy Chemistry, 25: 101-109, 2016.
4. In Supplementary Fig. S3. The authors indicated the spectra as XRD FTIR spectra. It should be FTIR.

Reviewer #2 (Remarks to the Author):

This is overall a very careful and detailed paper, which merits publication. However, there is perhaps too much laudatory language e.g., ultra-stable, drastically enhanced for my taste. The legends below some of the Figures are minor essays in themselves and take a great deal of studying, perhaps this cannot be avoided but it makes the paper more difficult to read. Also, the authors also need to check their English for minor faults, and make sure their references are in consistent format. The authors need to check their references both in the main text and the SI file.

Minor

Pg. 1 the authors mention CLC reactors but reference only a 1 MWth German plant, can they reference any other larger pilot scale units?

Pg. 2 decide on coprecipitation or co-precipitation on page 3.

Don't define acronyms multiple times, e.g. page 3 and 7 for TGA. Also, make sure you define acronyms at the first point of use, e.g. Fig. 2 on page 4 is the first appearance of STEM

Pg. 6 Using concentrated NaOH ... define concentrated in this context.

Pg. 7 be consistent e.g. calcined at 950°C for 3 h but on page 5 t 950 °C for 3 hours

FBR first appears on page 10 and is defined on page 15, should you say something about your reactor in the main text, maybe even the size?

Pg. 14 s. For chemical looping processes using fluidized bed reactors, attribution of particles is one of the most critical obstacles that limit the commercialization. I presume you don't actually mean attribution.

Pg. 14 was prepared by a precipitating agent of NaOH

Pg. 14. However, effect of residual sodium impurity on the structure of electrocatalysts and electrochemical reactions was poorly understood.. English structure.

Pg. 15 What is the difference between SATP and STP?

Why does reference 11 have no volume or page numbers when I look at the paper itself it has both.

Ref 32, 3, 34, 35,40 subscript chemical symbols even in the reference section. Also avoid oddities like: CO₂capture.

Why Progress in Energy and Combustion Science but an ampersand for EES?

Reviewer #3 (Remarks to the Author):

This manuscript describes the preparation and characterization of high-performance oxygen storage materials for applications in chemical looping processes. The authors demonstrated that the synthesis route with CuMgAl hydrotalcite precursors led to highly-dispersed CuO nanoparticles on a stable Mg-Al oxide support. The resultant copper-based materials exhibited enhanced reaction rates and stable O₂ storage capacity in repeated redox cycles. The manuscript is well written, and the findings reported herein are significant in terms of chemical looping technologies and materials engineering. I thus believe that the present work attracts broad interest in the relevant research community and deserves publishing in a high-impact journal.

To improve the quality of the manuscript, I suggest the authors make minor revisions. Detailed comments are given below. With adequate revisions, I will recommend this manuscript to be accepted for publication.

(1) Whereas the authors emphasize the importance of "precursor chemistry," I feel that the description of the formation mechanism of the hydrotalcite precursors is not sufficient. In fact, the role of Na₂CO₃ as the precipitant is unclear, despite the experimental fact that the LDH precursor with the NaOH/Na₂CO₃ mixed precipitant is better than that prepared using NaOH solely. It would be inferred that the incorporation of Na₂CO₃ effectively suppresses rapid nucleation (due to its buffering effect?) and improves the chemical homogeneity. This point should be discussed more in detail.

(2) Page 9, redox cycling behaviors of the oxygen storage materials are demonstrated, and CuAl-MMO (Mg-free) is found to involve redox of copper aluminates $\text{CuAl}_2\text{O}_4/\text{CuAlO}_2$ (Figs. 5d and 5e). It seems that the redox of $\text{CuAl}_2\text{O}_4/\text{CuAlO}_2$ is not straightforward. The formation of Cu_2O in the reduced material cannot be assumed as a decomposition product of CuAl_2O_4 because CuAl_2O_4 is more Cu-deficient than its reduced form CuAlO_2 . The authors should discuss the reaction mechanism.

(3) Page 12, the efficient gas purification by redox cycling of CuMgAl-MMO is demonstrated. Figures 7b-d reveal that the CuMgAl-MMO material effectively oxidizes the incoming CO and CH₄ at a low temperature of 400C. To justify that CuMgAl-MMO is promising for CLC applications, the remanent CO/CH₄ concentrations in the purified argon flow should be indicated.

(4) Page 6, Caption of Figure 3 describes that (b) is a diagram for $\text{CuO}/\text{MgAl}_2\text{O}_4$ derived from low-crystallinity CuMgAl-LDH, and (d) from high-crystallinity LDH. It seems that the assignments are opposite: (b) and (d) should be derived from high-crystallinity and low-crystallinity CuMgAl-LDH, respectively. A similar mixed-up is also seen in the 2nd paragraph of Page 7.

(5) Several errors in Main text and Supplementary should be corrected.

[Main text]

Page 8, Line 5. "Fig. S13" should be corrected as "Figs. S11, S12."

Page 10, the 8th line from the bottom. "Fig. S19" should be corrected as "Fig. S22."

Page 14, Line 5. "attribution" should be corrected as "attrition."

[Supplementary]

Page 8, Caption of Fig. S3. "XRD" at the beginning should be removed. In the 6th line of the caption, "a precipitant for (B)" should be corrected as "a precipitant for (A)."

Page 23, Caption of Fig. S17. "on the left" and "on the right" would be inappropriate. These words need to be replaced by, e.g., "on the top" and "on the bottom."

Page 25, Caption of Fig. S19. "by rich in Na" should be corrected as "be rich in Na."

Page 26, Caption of Fig. S20. The sentence "The needle-like... Na-Al-O oxides)." seems inappropriate and should be corrected (or removed).

The authors thank the reviewers for their valuable time and insights in reviewing and improving our manuscript. The reviewers have constructed many valid criticisms and we have changed the manuscript in several locations to improve the clarity and quality of our work by incorporating reviewers' suggestions.

Below we include a detailed response to each reviewer's comments.

Response to Reviewer #1

- 1) **The novelty of this manuscript is relatively low considering the CuO/MgAl₂O₄ oxygen carrier has been extensively applied to chemical looping combustion in the context of the oxygen uncoupling scheme for combustion (recent examples: Kuang et al, Energy & Fuels, 35(1): 618-625, 2020; Zheng et al, Energy & Fuels, 34(7): 8718-8725, 2020).**

Response: We thank the reviewer for their comments and would like to address the reviewer's concerns. We believe the novelty of the present work has not been fully recognised. We did not claim that CuO/MgAl₂O₄ as oxygen carriers for chemical looping combustion is the novelty of our work. In fact CuO/MgAl₂O₄ have been prepared by conventional methods, for example, spray drying of 60wt% CuO and 40wt% MgAl₂O₄. The authors (Energy Fuels 2012, 26, 5, 3069–3081) concluded "*Nevertheless, after 40 h of continuous operation at high temperatures, the particle integrity decreased significantly, indicating the need to improve the lifetime of this kind of material for use in an industrial CLOU process.*". This confirms that more innovative preparation methods are desirable to generate materials that meet the criteria for use in chemical looping process.

Adánez-Rubio, et al, Evaluation of a Spray-Dried CuO/MgAl₂O₄ Oxygen Carrier for the Chemical Looping with Oxygen Uncoupling Process. Energy Fuels 2012, 26, 5, 3069–3081

Prospects of Al₂O₃ and MgAl₂O₄-Supported CuO Oxygen Carriers in Chemical-Looping Combustion (CLC) and Chemical-Looping with Oxygen Uncoupling (CLOU). Energy Fuels 2011, 25, 11, 5493–5502.

Furthermore, we want to emphasise that the examples provided by the reviewer did not study CuO/MgAl₂O₄ oxygen carriers. Kuang et al, Energy & Fuels, 35(1): 618-625, 2021 investigated unsupported CuO for coal combustion and concluded "*However, as we all know, CuO easily suffers from sintering and agglomeration at high temperatures. A higher temperature leads to a higher reaction rate and power generation efficiency. Modification of the support is a good choice to avoid the deactivation of CuO in iG-CLC and iG-CLOU processes*". The second example, Zheng et al, Energy & Fuels, 34(7): 8718-8725, 2020, also investigates unsupported CuO and focus on the oxidation mechanism of CuO-Cu₂O-Cu. Both examples studied oxygen carriers in thermogravimetric analysers and performed very limited material characterisation.

These previous studies support the importance of using a design strategy (such as the LDH-MMO synthesis route outlined in the present work) to produce Cu-based oxygen carriers that do not suffer from sintering and agglomeration at high temperatures, *i.e.* these oxygen carriers consisting of a highly dispersed CuO phase within a highly inert MgAl₂O₄ support.

We need to emphasize that the novelty of our work lies in the development and optimisation of materials chemistry of layered double hydroxide precursors and derived mixed metal oxides. We rigorously characterise their precursors as well as freshly calcined and redox-cycled materials and elucidate reaction mechanisms during synthesis and during redox-experiments. The suggested synthesis procedure effectively inhibits contamination of sodium-species, thereby resolving a problem that

occurred in previous LDH works. We demonstrate how tuning the precursor LDH chemistry can produce oxygen carrier or redox sorbents with optimised performance and remarkably high stability.

The effect of synthesis conditions on the structure of the LDH precursors, calcined MMOs, and cycled oxygen carriers was assessed through rigorous characterisation, which (to our best knowledge) was not carried out in the literature. The durability of oxygen carriers in this work were tested in a fluidised bed reactor, a system in which the oxygen carriers are also exposed to mechanical forces, and detailed characterisation of the LDH precursors, and fresh and cycled MMOs was carried out. We thus believe our results represents a significant advancement in the field and hope it will inspire the wider chemical looping research community to adopt the LDH-MMO design strategy to improve the performance of oxygen carriers for conventional chemical looping systems and beyond combustion.

2) Additionally, the nanostructure in precursors, which was claimed as a key feature of this work, is very common in co-precipitation synthesis of materials.

Response: We do not claim that the layered structures are novel features either. The layered features are easily formed during coprecipitation (e.g. metal hydroxides, or clay materials) but often the materials chemistry is poorly understood and the precursors are insufficiently characterised. Furthermore, the aim of this work was not to produce novel nanostructures, but to use the LDH-synthesis route to produce oxygen storage materials with a high degree of atomic dispersion of metals in the LDH structure (which in turn leads to a high resistance to mechanical and chemical stresses and a high selectivity in the formation of the desired active and supporting phases). The precursor chemistry of co-precipitated oxygen carriers is generally poorly understood or not investigated. In the present work, we tuned the synthesis conditions to optimise the nanostructure of the LDH precursor to further improve the mechanical strength of the resulting oxides, while we show that we can retaining a high degree of dispersion of the active material within the support.

3) The authors were unable to reveal the reaction chemistry of or the mechanism underlining how Na and Mg ions inhibit CuAl_2O_4 formation, which is the key to and the most interesting part of this work. As such, this work is premature and not recommended for publication in a premier journal such as Nature Communication.

Response: We agree the inhibition of CuAl_2O_4 formation by Mg is a key part of this work and has been outlined in other works, but no mechanism for inhibition was proposed in these works (C. R. Forero, *International Journal of Greenhouse Gas Control*, 5: 659–667, 2011; P. Gayán et al, *Energy & Fuels*, 25: 1316–1326, 2011). To demonstrate the inhibiting effect of Mg on active phase–support interaction, we have added sections to the manuscript and to the supplementary information. For these additional parts, we have carried out thermodynamic calculations with FactSage (Bale *et al.*, *Calphad*, 54: 35-55, 2016) for the calcined system in the presence and absence of Mg. This has been done for 600 °C and 1000 °C (and temperatures in-between) to cover a wide range of potential operating temperatures for CLC. We explain the mechanism of how the Mg ions inhibit the undesirable Cu-Al spinel formation, amongst others, by plotting the equilibrium composition as a function of the oxygen concentration, since the solid-state reaction between the active phase and the support for the Cu-Al-O system (which is prone to these interactions) is thermodynamically unfavourable in the fully oxidised state (Figure S6A and S6B with x in $\text{Cu} + 2 \text{Al} + \text{Mg} (0) + x \text{O}_2 = 2.0$). In the fully reduced state ($x = 1.5$) Cu and MgAl_2O_4 are also thermodynamically stable. However, at intermediate oxidation states or upon O_2 removal during thermal decomposition of CuO, the Gibbs free energies of the species allow for formation of Cu-Al oxides. We have discussed this phenomenon as well as observations reported in the literature, in the main article.

For the trimetallic-system Cu-Al-Mg-O (Figure S6C and S6D), interactions between the active phase and the supporting phase are thermodynamic not favourable in the temperature range 600–1000 °C across the entire oxidation state range, as shown in Figure S6C and S6D. This has also been reported by Takaya *et al.*, ACS Earth and Space Chemistry, 3: 285-294, 2019.

The use of sodium to inhibit the formation of CuAl₂O₄ was discussed in Song *et al.*, Energy & Environmental Science, 6: 288-298, 2013. The sodium originating from the co-precipitation of the LDH materials was suggested to prevent the formation of CuAl₂O₄ by limiting the interaction of Al₂O₃ with copper atoms during calcination through the formation of sodium aluminates. Similarly, El-Shobaky, et al., Thermochimica Acta, 141: 195–203, 1989 theorised the formation of sodium aluminates at CuO-Al₂O₃ grain boundaries inhibit the diffusion of Cu⁺ and Cu²⁺ ions into the Al₂O₃ matrix to form CuAl₂O₄ and CuAlO₂. The formation of dawsonite has also been carefully studied (Geochimica et Cosmochimica Acta 71 (2007) 4438–4455).

Figure R1. (A) Phase diagram of precipitation of dawsonite/boehmite, and associated reactions (a-f) (Geochimica et Cosmochimica Acta 71 (2007) 4438–4455). (B) XRD patterns of (A) aluminium hydroxides precursor and (B) calcined product prepared with varied alkaline solution concentrations at constant pH (9.5-10.0). Samples in (B) and (C): (a) 1 M NaOH, (b) 0.94 M NaOH+0.06 M Na₂CO₃, (c) 0.5 M NaOH+0.5 M Na₂CO₃, (d) 1 M Na₂CO₃. AlOOH (pseudoboehmite, JCPDS 49-0133), NaAlCO₃(OH)₂ (Dawsonite, JCPDS 45-1359), Al₂O₃ (JCPDS 10-0425), NaAlO₂ (JSPDS, 33-1200). Results adapted from the previous study (Energy Environ. Sci., 2013,6, 288-298).

References:

Dawsonite synthesis and reevaluation of its thermodynamic properties from solubility measurements: Implications for mineral trapping of CO₂. Geochimica et Cosmochimica Acta 71 (2007) 4438–4455

We have added the discussion of the inhibition of CuAl₂O₄ formation by Mg to the supplementary information.

Supplementary Fig. S6. Plots of the thermodynamic equilibrium composition as a function of the oxygen concentration using data from FactSage for the systems (A, B) Cu-Al-O and (C, D) Cu-Al-Mg-O, and the temperatures (A, C) 600 °C and (B, D) 1000 °C.

4) **The nanostructures in the precursors were not investigated properly. The STEM images in Fig 2 show amorphous structure with no obvious lattice fringes, which is contradictory to the XRD observation that the oxygen carrier is highly crystalline.**

Response: We have performed extensive characterization analyses to investigate the structure and properties of the precursors. We performed additional STEM imaging of LDH precursor, confirming the formation of typical nanosheets-like structures. EDX analysis confirm the homogeneous distribution of the elements. We have also tried HR-TEM of the LDH precursors, however, the decomposition of the LDH precursor under high electron beam caused problems to obtained HR-TEM imaging data. Nevertheless, the HAADF-STEM confirmed the high crystallinity of the precursor.

Figure R2. HAADF-STEM, STEM, and EDX analysis of CuMgAl-LDH precursor.

Figure R3. STEM image of CuMgAl-MMO calcined at 800°C (cooled down to room temperature immediately).

Figure R4. STEM image of CuMgAl-MMO calcined at 800°C (maintained for 3h).

The reviewer also quired about the amorphous structure of STEM of calcined mixed metal oxides. The original figure presented in the manuscript was probably not clear due to low resolution PDF file. As shown above in Fig. R4, the sample calcined at 800°C for short period of 3h show nanoscale crystals observed at high magnification HR-TEM image. The fast Fourier transform (FFT) patterns of different regions clearly suggest the formation of crystalline domains, however specific assignment of crystalline phases could not be obtained, which might be due to defects in the Cu phases, interactions between Cu and Mg-Al oxide support at the interfaces or partial incorporation of Cu into the spinel support.

We have updated these results in the main figure 2, and moved some STEM images to supplementary information Fig. S8.

5) This work was flooded with SEM images, which is very unnecessary. No detailed nanostructure information was provided.

Response: We thank the reviewer for their suggestion. However, we believe that the SEM images presented in the main text are necessary to show the morphology transition from typical LDH morphology to MMO after calcination, and the further morphology evolution upon redox-cycling. The micrographs also demonstrate that the structural integrity of the particles is maintained over long-term cycling at high temperatures.

The SEM images supplied in the supplementary information show the detailed morphology change for LDHs with different compositions (*e.g.* for increasing mol% of Mg) as well as for samples prepared by different synthesis conditions and co-precipitation agents (NaOH, NaOH + Na₂CO₃). These SEM images have been included in the supplementary information to provide a deeper insight into the effect of composition and synthesis conditions on the material structure, which may provide relevant insights for readers that work outside of the field of chemical looping. The morphologies of these materials differ significantly and further complimentary characterisation data (including room temperature and *ex-situ* high temperature XRD and elemental analyses) were provided in the supplementary information to allow the readers to draw further conclusions.

6) The redox cycles in this work has been tested up to 100 cycles in TGA and 20 cycles in FBR. This is significantly less compared with the state-of-the-art chemical looping systems that use CuO/MgAl₂O₄ as oxygen carriers, such as Zheng, et al, Journal of Energy Chemistry, 25: 101-109, 2016.

Response: The reviewer has concern about the number of redox cycles. We have performed additional experiments to address this concern. It should be noted that the oxygen carriers tested by Zheng, *et al.*, Journal of Energy Chemistry, 25: 101-109, 2016 were **CuO/SiO₂ prepared with a mechanical mixing method instead of CuO/MgAl₂O₄**. The Cu/SiO₂ materials were cycled for 420 cycles in a fixed bed reactor at low temperature of 500°C. ***The material exhibited high redox reactivity in the initial cycles but degradation occurred with cycle number, which was attributed to fragmentation of secondary particles and local agglomeration of fine particles.***

A fixed bed reactor system was used in Zheng, *et al.*, Journal of Energy Chemistry, 25: 101-109, 2016, whereas FBR systems are likely to be the reactor type of choice for scaled-up processes. A FBR system was used in this work which importantly provides greater insight than fixed bed reactors with respect to the resistance to mechanical stresses. In the present work, the oxygen carriers were cycled for 100 cycles in a FBR to assess the CLOU performance, and 20 cycles in a FBR to assess the CLPO performance. We believe 100 cycles in a FBR are sufficient to screen oxygen carriers for further

development for the use in scaled-up systems. However, we have performed an additional experiment to address the issue raised by the reviewer. The CuMgAl-MMO was cycled for 500 CLOU cycles in a TGA at 900 °C. The TGA can be considered to be a variant of a classical fixed bed reactor with a low heating rate, and the experiments are thus similar in nature to those performed by Zheng, *et al.* (Journal of Energy Chemistry, 25: 101-109, 2016). The results of the additional TGA experiment showed the oxygen release capacity of the CuMgAl-MMO-HS was maintained over 500 CLOU cycles at 900 °C. Correspondingly, we have revised the text in the manuscript and have added the following figure to the supplementary information:

Supplementary Fig. S26. TGA profiles of long-term redox cycling of CuMgAl-MMO-HS. (a) Cyclic O₂ release and storage at 900°C for 500 cycles, phase change between CuO and Cu₂O. (b) O₂ release capacity over 500 cycles of O₂ release.

7) In Supplementary Fig. S3. The authors indicated the spectra as XRD FTIR spectra. It should be FTIR.

Response: We thank the reviewer for carefully reviewing this manuscript. This typo has been corrected.

Response to Reviewer #2

This is overall a very careful and detailed paper, which merits publication. However, there is perhaps too much laudatory language e.g., ultra-stable, drastically enhanced for my taste. The legends below some of the Figures are minor essays in themselves and take a great deal of studying, perhaps this cannot be avoided but it makes the paper more difficult to read. Also, the authors also need to check their English for minor faults, and make sure their references are in consistent format. The authors need to check their references both in the main text and the SI file.

Response: We thank the reviewer for their positive comments. We have improved the language, and have shortened/streamlined sections of the main body and figure captions. The references in the main text and supplementary information have also been checked and corrected. We thank the reviewer for their diligence in identifying errors in the manuscript.

Minor

Pg. 1 the authors mention CLC reactors but reference only a 1 MWth German plant, can they reference any other larger pilot scale units?

Response: We have added additional references of larger-scale pilots, including 3MWth prototype developed by Alstom in the USA, pilot-scale plants being developed in China, and a latest paper by Chalmers Group in Sweden on design proposal for a 200 MWth Chemical looping CFB reactor.

References:

Andrus, H. E. Alstom's Chemical Looping Combustion Technology for CO₂ Capture for New and Retrofit Coal-Fired Power Plants. United States: N. p., 2017. Web. doi:10.2172/1440120.

Zhao, H. et al, Chemical Looping Combustion of Coal in China: Comprehensive Progress, Remaining Challenges, and Potential Opportunities. Energy Fuels 2020, 34, 6, 6696–6734.

Lyngfelt, A. et al, Achieving Adequate Circulation in Chemical Looping Combustion—Design Proposal for a 200 MWth Chemical Looping Combustion Circulating Fluidized Bed Boiler. 2022. Energy Fuels, <https://doi.org/10.1021/acs.energyfuels.1c03615>.

Pg. 2 decide on coprecipitation or co-precipitation on page 3.

Response: This minor mistake has been corrected to co-precipitation.

Don't define acronyms multiple times, e.g. page 3 and 7 for TGA. Also, make sure you define acronyms at the first point of use, e.g. Fig. 2 on page 4 is the first appearance of STEM

Response: This mistake has been corrected, acronyms are now defined once and at the first point of use.

Pg. 6 Using concentrated NaOH ... define concentrated in this context.

Response: This minor mistake has been corrected / revised as: "using 2 M NaOH"

Pg. 7 be consistent e.g. calcined at 950°C for 3 h but on page 5 t 950 °C for 3 hours

Response: We have eliminated these inconsistencies.

FBR first appears on page 10 and is defined on page 15, should you say something about your reactor in the main text, maybe even the size?

Response: We thank the reviewer for their suggestion, a reference to a paper detailing the FBR system has been added to the experimental section and we provide a drawing of the set-up and further information in the supplementary information. The size is also added in the experimental section.

Pg. 14 s. For chemical looping processes using fluidized bed reactors, attribution of particles is one of the most critical obstacles that limit the commercialization. I presume you don't actually mean attribution.

Response: This spelling error has been corrected: "For chemical looping processes using FBRs, attrition of particles is one of the most critical obstacles that limit commercialisation."

Pg. 14 was prepared by a precipitating agent of NaOH

Response: This mistake has been corrected: "The CuAl-LDH was prepared using NaOH and Na₂CO₃ as precipitating agents"

Pg. 14. However, effect of residual sodium impurity on the structure of electrocatalysts and electrochemical reactions was poorly understood. English structure.

Response: This minor mistake has been corrected / revised as: "However, the effect of the residual sodium impurity on the structure of electrocatalysts and electrochemical reactions was poorly understood."

Pg. 15 What is the difference between SATP and STP?

Response: This minor mistake has been corrected to SATP and the abbreviation was introduced properly: "standard atmospheric temperature and pressure (SATP, *i.e.* 20 °C and 101.3 kPa)"

Why does reference 11 have no volume or page numbers when I look at the paper itself it has both.

Response: The reference has been corrected to contain now the volume and page numbers.

Ref 332, 3, 34, 35,40 subscript chemical symbols even in the reference section. Also avoid oddities like: CO₂capture.

Response: These references have been corrected and oddities have been removed.

Why Progress in Energy and Combustion Science but an ampersand for EES?

Response: The EES references have been corrected.

Response to Reviewer #3

This manuscript describes the preparation and characterization of high-performance oxygen storage materials for applications in chemical looping processes. The authors demonstrated that the synthesis route with CuMgAl hydrotalcite precursors led to highly-dispersed CuO nanoparticles on a stable Mg-Al oxide support. The resultant copper-based materials exhibited enhanced reaction rates and stable O₂ storage capacity in repeated redox cycles. The manuscript is well written, and the findings reported herein are significant in terms of chemical looping technologies and materials engineering. I thus believe that the present work attracts broad interest in the relevant research community and deserves publishing in a high-impact journal.

Response: We thank the reviewer for their positive feedback and providing insightful comments to further improve the quality of our manuscript.

To improve the quality of the manuscript, I suggest the authors make minor revisions. Detailed comments are given below. With adequate revisions, I will recommend this manuscript to be accepted for publication.

1) Whereas the authors emphasize the importance of “precursor chemistry,” I feel that the description of the formation mechanism of the hydrotalcite precursors is not sufficient.

Response: We thank the reviewer for highlighting this weakness. We have carried out a thermodynamic calculation (Supplementary Figs. S5) to elucidate mechanisms and identify all (possible) intermediates and products in the precursor synthesis (*i.e.* during co-precipitation). We have added a comprehensive discussion (3 pages long) to the supplementary information on this topic. As replied to Reviewer 1, we have added the discussion on the formation of the dawsonite. In brief, in an aqueous Na-Cu-Al system, dawsonite appears to be stable over the *pH* value range of interest in the absence of Mg. However, when adding Mg to the system, the dawsonite log concentration falls to below -3.0 for the *pH* value range of interest. Mg combines with carbonate groups and Al species thereby removes all Al required for dawsonite formation. We have added the following Figure to the supplementary information:

Supplementary Fig. S5. Plots of equilibrium compositions (A) in the absence and (B) presence of Mg ions as a function of the pH value. The plots were generated with SPANA (formerly MEDUSA) using data from DATABASE (formerly HYDRA) ¹⁶ as well as literature values for the compounds listed in Table SX. The following ion concentrations were used: (A): $[\text{Na}^+] = [\text{Al}^{3+}] = [\text{CO}_3^{2-}] = 1 \text{ M}$ and (C): $[\text{Na}^+] = [\text{Mg}^{2+}] = [\text{Al}^{3+}] = [\text{CO}_3^{2-}] = 1 \text{ M}$.

2) In fact, the role of Na_2CO_3 as the precipitant is unclear, despite the experimental fact that the LDH precursor with the $\text{NaOH}/\text{Na}_2\text{CO}_3$ mixed precipitant is better than that prepared using NaOH solely. It would be inferred that the incorporation of Na_2CO_3 effectively suppresses rapid nucleation (due to its buffering effect?) and improves the chemical homogeneity. This point should be discussed more in detail.

Response: We thank the reviewer for raising an interesting query. At high concentrations of NaOH , the level of supersaturation is higher which results in higher rates of nucleation and growth is inhibited (Clark *et al.* Journal of Colloid and Interface Science, 504: 492-499, 2017). The addition of carbonate during precipitation has been observed to improve stacking in the c direction (perpendicular to the LDH sheet) due to the different charge balancing anions present in the interlayer (Kim *et al.*, Nanomaterials, 11(11): 2809, 2021; Tichit *et al.*, Chemical Engineering Journal, 369: 302-332, 2019). The LDH

precursor synthesised with 2 M NaOH as the precipitating agent produced LDHs with NO_3^- as the main interlayer anion (LDH- NO_3) with some CO_3^{2-} due to the strong affinity of the LDH for CO_3^{2-} and dissolution of ambient CO_2 into the reaction vessel, as has been reported elsewhere (Hunter et al., Energy & Environmental Science, 9:1734-1743, 2016). Whereas the interlayer anion was CO_3^{2-} (LDH- CO_3) when 1 M NaOH and 1 M Na_2CO_3 was used as the precipitating agent. The LDH- CO_3 and LDH- NO_3 represent the most stable and one of the least stable LDHs, respectively (Miyata, Clays and Clay Minerals, 31, 305-311, 1983), and the conversion of LDH- CO_3 into anion-exchangeable LDHs, such as LDH- NO_3 , is an important area of LDH research (Iyi & Yamada, Chemistry Letters, 39: 591-593, 2010). When all other synthesis conditions are kept constant, LDH- NO_3 have been observed to be less crystalline, less stable, more disordered, and form denser aggregates than LDH- CO_3 (Sun & Dey, Journal of Colloid and Interface Science, 48: 160-168, 2015). Sun & Dey found the smaller size of the LDH- NO_3 compared to LDH- CO_3 was due to the weaker electrostatic bonds between interlayer anions and double hydroxide layers of the former. The severe aggregation of the LDH- NO_3 was attributed to the lower crystallinity of the LDH resulting in lower charge density and enthalpy of hydration of the edges sites of the LDH where agglomeration occurred (Sun & Dey, Journal of Colloid and Interface Science, 48: 160-168, 2015).

We have revised the text in the main article accordingly as follows:

“The varied supersaturation rate and intercalated anion compositions and have critical effect on the formation of the LDH precursors, crystallization and growth^{47, 48}. Using 2 M NaOH as the precipitating agent produced a low-crystallinity LDH (Supplementary Fig. S2C) due to the rapid nucleation and precipitation of ultra-fine low-crystallinity nanoparticles in the precursor, as observed by SEM (Supplementary Fig. S9). Whereas high-crystallinity LDH was produced when using 1 M NaOH + 1 M Na_2CO_3 (Supplementary Fig. S2a) due to the low supersaturation and therefore lower rate of nucleation and increased growth. The use of Na_2CO_3 has also been observed to increase stacking in the *c*-direction, perpendicular to the LDH sheets, due to the stronger electrostatic bonds between CO_3^{2-} (*versus* NO_3^-) and the brucite layers. After calcination at high temperatures, the MMOs prepared using solely NaOH generally showed a lower surface area compared to those prepared using a mixture of NaOH and Na_2CO_3 (Supplementary Fig. S14 and Table S5)”

3) Page 9, redox cycling behaviors of the oxygen storage materials are demonstrated, and CuAl-MMO (Mg-free) is found to involve redox of copper aluminates $\text{CuAl}_2\text{O}_4/\text{CuAlO}_2$ (Figs. 5d and 5e). It seems that the redox of $\text{CuAl}_2\text{O}_4/\text{CuAlO}_2$ is not straightforward. The formation of Cu_2O in the reduced material cannot be assumed as a decomposition product of CuAl_2O_4 because CuAl_2O_4 is more Cu-deficient than its reduced form CuAlO_2 . The authors should discuss the reaction mechanism.

Response: We thank the reviewer for their comments and have investigated the thermodynamics of the Cu-Al-O system to improve the understanding of the oxygen storage and release reactions. A phase diagram was produced using the Gibbs free energy data reported by Jacob & Alcock, Journal of the American Ceramic Society, 58(5-6), 192-195, 1975. Calcination of the 1:1 Cu:Al LDH precursor in air at 950 °C produces an oxygen carrier of CuO supported on copper aluminate spinel (CuAl_2O_4), as confirmed by powder XRD analysis (Fig.5e, fresh material). During redox cycling at 900 °C the oxygen partial pressure is sufficiently low during reduction for the active cupric oxide (CuO) to decompose to cuprous oxide (Cu_2O) and oxygen through reaction (1),

The CuO could also react with CuAl_2O_4 to form a delafossite-type material (CuAlO_2) and oxygen through reaction (2),

The CuAl_2O_4 could also react to form CuAlO_2 and alumina (Al_2O_3) and oxygen through reaction (3),

Although the three reactions are thermodynamically feasible, the absence of CuAlO_2 in the XRD pattern of the O_2 released sample (Fig. 5e) and appearance in the cycled material (Fig. 5e) indicates reaction (2) was kinetically limited, as observed in Hu et al., RSC Advances, 6: 113016, 2016. The CuAlO_2 peak in the cycled material confirms some of the active CuO was not regenerated during re-oxidation, representing a loss in oxygen storage capacity over 100 cycles as seen in Fig. 5c. The oxidation of CuAlO_2 (reverse reaction (2)) is limited by kinetics and not thermodynamics, and therefore the active CuO may be recovered if the segment time was increased, however this would severely impact the operation and economics of commercial chemical looping units. The absence of Al_2O_3 from all XRD patterns (Fig. 5e) indicates reaction (3) did not proceed at an appreciable rate.

We have added this discussion of the oxygen release and storage reaction mechanism for the CuAl- MMO to the supplementary information. We have edited the text in the main article as follows:

“The oxygen release and storage reaction mechanisms of the CuAl- MMO are discussed in the supplementary information (Supplementary Fig. S21).”

Additionally, we have added the following figure to the supplementary information:

Supplementary Fig. S21. Cu-Al-O system equilibrium phase diagram for a 1:1 molar mixture of CuO and Al_2O_3 based on data reported by Jacob & Alcock. The cycling conditions used in this work (Fig. 5 and 6, Supplementary Fig. S18) are demarcated by the CLOU and oxidation regions.

(3) Page 12, the efficient gas purification by redox cycling of CuMgAl- MMO is demonstrated. Figures 7b-d reveal that the CuMgAl- MMO material effectively oxidizes the incoming CO and CH_4 at a low temperature of 400C. To justify that CuMgAl- MMO is promising for CLC applications, the remanent CO/CH_4 concentrations in the purified argon flow should be indicated.

Response: We thank the reviewer for their suggestion and have included additional figures in the supplementary information with magnified y-axes to show the changes in CO/CH_4 gas concentration

during CLPO. The residual CO/CH₄ concentrations are usually below detecting limit, which confirm their high conversion (close to 100%). The intrinsic accuracy of the measurement is 1.0% of reading, and the detection limits of the MGA3000C Multi-Gas Analyser (ADC Gas Analysis) are 500 ppm for CO, CO₂ and CH₄ and 1000 ppm for O₂ which have been indicated in the figure caption.

We have changed the text in the main article as follows: “Additional gas profiles can be found in the supplementary information (Supplementary Fig. S28, S29).”, and have added an additional figure to the supplementary information, which is reproduced below:

Supplementary Fig. S29. Enhanced gas concentration profiles of chemical looping combustive purification (CLCP) cycling of CuMgAl-MMO in FBR. (a) 2 vol% CO/N₂ used for the reduction and oxidation under 10 vol% O₂/N₂ at 400 °C, (b) 2 vol% CO/N₂ used for the reduction and oxidation under 10 vol% O₂/N₂ at 800 °C, (c) 1 vol% CH₄/N₂ used for the reduction and oxidation under 10 vol% O₂/N₂ at 800°C. The intrinsic accuracy of the measurement is 1.0% of reading, and the detection limits of the MGA3000C Multi-Gas Analyser (ADC Gas Analysis) are 500 ppm for CO, CO₂ and CH₄ and 1000 ppm for O₂.

- 4) Page 6, Caption of Figure 3 describes that (b) is a diagram for CuO/MgAl₂O₄ derived from low-crystallinity CuMgAl-LDH, and (d) from high-crystallinity LDH. It seems that the assignments are opposite: (b) and (d) should be derived from high-crystallinity and low-crystallinity CuMgAl-LDH, respectively. A similar mixed-up is also seen in the 2nd paragraph of Page 7.

Response: We thank the reviewer for their comment. We need to explain that the original results in Figure 3 were correct. The morphology of the mixed metal oxides is related to the structure of the precursors. Low-crystallinity CuMgAl-LDH tend to aggregate and sinter to form relatively larger CuO particles. From Fig. S2, the XRD peaks for the LDH materials synthesised using 1 M NaOH + 1 M Na₂CO₃ (Fig. S2A and B) are much sharper (high crystallinity) than the broad peaks observed for the low crystallinity LDH material synthesised with 2 M NaOH (Fig. S2C and D). As mentioned in response to comment (1), the difference in crystallinity is due to the higher concentration of NaOH, and hence higher rate of nucleation rate of the LDHs in the 2 M NaOH materials. Upon high temperature calcination, the layered structure of the high crystallinity CuMgAl-LDH materials are retained and the spatial confinement of atoms reduces the extent of sintering (Fig 3d). Whereas the low-crystallinity LDH structure is lost at lower calcination temperatures, and distinct aggregation and sintering is more clearly observed in the STEM imaging and energy-dispersive X-ray elemental mapping (Fig. 3b).

Similarly, the

The text in the main article was changed to the following:

“The varied supersaturation rate and intercalated anion compositions and have critical effect on the formation of the LDH precursors, crystallization and growth^{47,48}. Using 2 M NaOH as the precipitating agent produced a low-crystallinity LDH (Supplementary Fig. S2C) due to the rapid nucleation and precipitation of ultra-fine low-crystallinity nanoparticles in the precursor, as observed by SEM (Supplementary Fig. S9). Whereas high-crystallinity LDH was produced when using 1 M NaOH + 1 M Na₂CO₃ (Supplementary Fig. S2a) due to the low supersaturation and therefore lower rate of nucleation and increased growth. The use of Na₂CO₃ has also been observed to increase stacking in the *c*-direction, perpendicular to the LDH sheets, due to the stronger electrostatic bonds between CO₃²⁻ (*versus* NO₃⁻) and the brucite layers. After calcination at high temperatures, the MMOs prepared using solely NaOH generally showed a lower surface area compared to those prepared using a mixture of NaOH and Na₂CO₃ (Supplementary Fig. S14 and Table S5)”

- 5) Several errors in Main text and Supplementary should be corrected.

[Main text]

- Page 8, Line 5. “Fig. S13” should be corrected as “Figs. S11, S12.”
- Page 10, the 8th line from the bottom. “Fig. S19” should be corrected as “Fig. S22.”
- Page 14, Line 5. “attribution” should be corrected as “attrition.”

[Supplementary]

- Page 8, Caption of Fig. S3. “XRD” at the beginning should be removed. In the 6th line of the caption, “a precipitant for (B)” should be corrected as “a precipitant for (A).”
- Page 23, Caption of Fig. S17. “on the left” and “on the right” would be inappropriate. These words need to be replaced by, e.g., “on the top” and “on the bottom.”
- Page 25, Caption of Fig. S19. “by rich in Na” should be corrected as “be rich in Na.”
- Page 26, Caption of Fig. S20. The sentence “The needle-like... Na-Al-O oxides)” seems inappropriate and should be corrected (or removed).

Response: We thank the reviewer for carefully reviewing the text of the manuscript. We have resolved the issues as suggested and have improved the language and the spelling of the manuscript.

REVIEWERS' COMMENTS

Reviewer #1 (Remarks to the Author):

The authors added a very important section of the inhibiting effect of Mg on active phase–support interaction, which enhances the significance of this manuscript. Although the novelty of this work is not high, the fundamental approaches are sound. The manuscript is acceptable now.

Reviewer #3 (Remarks to the Author):

I reviewed the manuscript NCOMMS-21-47225A, the revised version of NCOMMS-21-47225 submitted last year by High et al. I found that the quality of the manuscript has been improved largely. The authors have adequately argued the concerns raised by Reviewer #1. Also, all of the issues pointed out by Reviewers #2, #3 have been addressed. I thus conclude that the present work is now substantially suitable for publishing in Nature Communications.

To further improve the quality of the manuscript, I encourage the authors to make minor revisions. In fact, both the main text and Supplementary Information still contain a lot of editorial errors (typos, wrong numbering, etc.). For details, please see the following comments. With appropriate revisions, I will recommend the present work to be accepted for publication.

[Main text]

Page 4,

9th line from the bottom. “purities” should be corrected to “impurities.”

8th line from the bottom. “We have added a discussion...” would be better to be rephrased by “A discussion on... has been given in the Supplementary Information.”

Page 6, 12th, 16th lines. “Cu-phases” seems to be an unclear term. It would be better to be replaced by “CuO particles.”

Page 7, 3rd line from the bottom. “pH” should not be written in italic.

Page 14, 1st line of Discussion. “have” should be corrected to “of.”

[Supplementary Information]

Page 10, 3rd line. “LDO” should be defined. Would it just be a typo of “LDH”?
“pH” should not be written in italic.

Page 11,

4th-5th lines. Chemical formulae should not be written in italic.

4th line. The table number “SX” should be corrected.

7th line. The figure number "1B" should be corrected to "S5B."

Page 12,

11th line. The figure number "1C" should be corrected to "S5C."

3rd line from the bottom. "In lab experiments" should be removed.

Page 13,

Fig. S6. The titles of Figs. A and B seem inappropriate. These figures are diagrams for the Mg-free system.

3rd line from the bottom. "... are thermodynamic not favorable" should be corrected to "... are thermodynamically unfavorable."

Page 25, 4th line. "(b) cyclic" should be corrected to "Cyclic."

Page 28. The numbering of equations should be corrected.

REVIEWERS' COMMENTS

Reviewer #1 (Remarks to the Author):

The authors added a very important section of the inhibiting effect of Mg on active phase–support interaction, which enhances the significance of this manuscript. Although the novelty of this work is not high, the fundamental approaches are sound. The manuscript is acceptable now.

Response: We thank the reviewer for valuable comments that improved the quality of our work.

Reviewer #3 (Remarks to the Author):

I reviewed the manuscript NCOMMS-21-47225A, the revised version of NCOMMS-21-47225 submitted last year by High et al. I found that the quality of the manuscript has been improved largely. The authors have adequately argued the concerns raised by Reviewer #1. Also, all of the issues pointed out by Reviewers #2, #3 have been addressed. I thus conclude that the present work is now substantially suitable for publishing in Nature Communications.

To further improve the quality of the manuscript, I encourage the authors to make minor revisions. In fact, both the main text and Supplementary Information still contain a lot of editorial errors (typos, wrong numbering, etc.). For details, please see the following comments. With appropriate revisions, I will recommend the present work to be accepted for publication.

[Main text]

Page 4,

9th line from the bottom. “purities” should be corrected to “impurities.”

8th line from the bottom. “We have added a discussion...” would be better to be rephrased by “A discussion on... has been given in the Supplementary Information.”

Page 6, 12th, 16th lines. “Cu-phases” seems to be an unclear term. It would be better to be replaced by “CuO particles.”

Page 7, 3rd line from the bottom. “pH” should not be written in italic.

Page 14, 1st line of Discussion. “have” should be corrected to “of.”

[Supplementary Information]

Page 10, 3rd line. “LDO” should be defined. Would it just be a typo of “LDH”?

“pH” should not be written in italic.

Page 11,

4th-5th lines. Chemical formulae should not be written in italic.

4th line. The table number “SX” should be corrected.

7th line. The figure number “1B” should be corrected to “S5B.”

Page 12,

11th line. The figure number “1C” should be corrected to “S5C.”

3rd line from the bottom. “In lab experiments” should be removed.

Page 13,

Fig. S6. The titles of Figs. A and B seem inappropriate. These figures are diagrams for the Mg-free system.

3rd line from the bottom. “... are thermodynamic not favorable” should be corrected to “... are thermodynamically unfavorable.”

Page 25, 4th line. “(b) cyclic” should be corrected to “Cyclic.”

Page 28. The numbering of equations should be corrected.

Response: We gratefully thank the reviewer for valuable feedback. We have revised these typos in the manuscript and supplementary information.